# Bridging On-Device and Cloud LLMs for Collaborative Reasoning: A Unified Methodology for Local Routing and Post-Training

**Wenzhi Fang** [1]   **Dong-Jun Han** [2]   **Liangqi Yuan** [1]   **Evan Chen** [1]   **Christopher G. Brinton** [1]

## Abstract

Device-cloud collaboration holds promise for deploying large language models (LLMs), leveraging lightweight on-device models for efficiency while relying on powerful cloud models for superior reasoning. A central challenge in this setting is determining, for each incoming query, whether it should be processed locally or offloaded to the cloud. Existing approaches typically rely on external routers, which often struggle to determine difficulty from the prompt itself, especially for tasks involving complex reasoning. Motivated by this limitation, we propose enabling on-device LLMs to decide internally whether to invoke cloud assistance at inference time, with this capability instilled through reinforcement learning based post-training. Casting on-device LLM post-training as a reward maximization problem, we design hierarchical rewards to encourage local problem solving and judicious cloud offloading. To solve the resulting problem, we develop an algorithm featuring a group-level policy gradient that stabilizes optimization, together with adaptive prompt filtering that provides complementary learning signals to mitigate policy collapse (i.e., exclusive local execution or exclusive cloud offloading). Extensive experiments on on-device-scale LLaMA and Qwen models across multiple reasoning benchmarks show that our method consistently outperforms baselines and significantly narrows the gap to full cloud LLMs.

## 1. Introduction

Large language models (LLMs) have achieved broad adoption in practice due to their strong performance (Wei et al., 2021; Touvron et al., 2023; Achiam et al., 2023). In typical cloud-based deployment settings, user queries are transmitted to powerful LLMs hosted on remote servers for processing. While effective, this paradigm imposes substantial computational pressure on cloud infrastructure (passed down to users through monetary cost), introduces non-negligible communication latency, and fails to fully leverage the potential of local computational resources (Jin & Wu, 2024). Given these issues, recent research has explored on-device LLMs, i.e., lightweight models optimized for deployment on local devices (Liu et al., 2024; Xu et al., 2024; Fang et al., 2025). However, due to their limited parameter volume, these on-device models often lag behind cloud LLMs in terms of performance, especially for tasks requiring sophisticated reasoning. This creates a critical trade-off between usage efficiency and task performance.

To overcome this efficiency-performance trade-off, recent approaches have aimed to facilitate device-cloud collaboration (Li et al., 2025). Within these frameworks, a local router, typically implemented as a separate classifier, decides whether a request should be processed by the on-device LLM or offloaded to the cloud (Ong et al., 2025), aiming to leverage the efficiency of the local model while retaining access to the superior performance of the cloud LLM. However, such routers require extra task-specific training after the on-device LLM's fine-tuning and and make routing decisions solely based on the prompt, which limits generalization and adaptability across tasks. Additionally, for reasoning tasks, the "surface-level" prompt features available to the local router often fail to reflect the underlying complexity of the problem. Moreover, treating the routing and the on-device LLM optimization as two decoupled processes ignores the potential of the model to jointly optimize its own problem-solving ability and collaboration, likely resulting in a suboptimal balance between on-device and cloud resource utilization. Motivated by this, we therefore pose the following two-fold question:

*1) Can the on-device LLM be trained to autonomously decide when to invoke the cloud LLM at inference time according to resource constraints? 2) Can this routing ability be jointly learned with reasoning (i.e., problem-solving) ability during the same post-training stage?*

---

[1]Purdue University [2]Yonsei University. Correspondence to: Wenzhi Fang <fang375@purdue.edu>.

*Proceedings of the 43rd International Conference on Machine Learning*, Seoul, South Korea. PMLR 306, 2026. Copyright 2026 by the author(s).

**Challenges.** Enabling on-device LLMs to autonomously decide when to invoke the cloud LLM at inference time eliminate the need for an external router, yet achieving this poses several challenges. First, reasoning and routing represent fundamentally different learning objectives. Second, jointly optimizing routing and the model's reasoning performance introduces a coupled learning challenge, as the on-device LLM must simultaneously improve its problem-solving ability while learning to judiciously invoke cloud assistance. Third, device-cloud collaboration mechanisms are typically constrained by an offloading budget, making it challenging to balance cloud calls with local computation. Overall, a principled approach that jointly optimizes problem-solving and routing under offloading constraints within a unified post-training framework remains elusive.

### 1.1. Contributions

Motivated by these observations, we propose a unified RL-based methodology that enables the on-device LLM to both strengthen its own reasoning for problem solving and learn routing strategies conditioned on its reasoning competence, by integrating routing optimization directly into post-training. Concretely, we cast this as a hierarchical reward maximization problem that assigns distinct rewards to different response types, including correct answers and calls for assistance, while constraining cloud LLM usage to mitigate over-reliance and promote efficient use of device resources. To solve this problem, we develop a Group-Adaptive Policy Gradient (GAPG) algorithm featuring (i) a group-level policy gradient and (ii) adaptive prompt filtering. Overall, we make the following contributions:

- *Unified formulation of collaborative reasoning.* We formulate a reward maximization problem with a theoretically inspired, collaboration-aware hierarchical reward design. Our formulation integrates routing optimization into the RL-based post-training, enabling the on-device model to simultaneously enhance its reasoning ability and discover informed routing strategies.

- *Group-adaptive policy gradient algorithm.* We propose a group-adaptive policy gradient algorithm with two key features: (i) group-level policy gradient that provably stabilizes the optimization, and (ii) adaptive prompt filtering that provides complementary learning signals for both local problem solving and assistance invocation. The filtering reduces the risk of policy collapse toward either exclusive local execution or exclusive cloud offloading.

- *Experimental validation.* Through extensive experiments across diverse models and benchmarks, we demonstrated that our approach consistently outperforms baselines in task tuning and routing, maintains stable training, and significantly narrows the performance gap to a full cloud LLM while adhering to call-for-cloud usage constraints.

### 1.2. Related Works

**Device-Cloud LLM Collaboration.** A growing body of work studies collaboration between resource-constrained device LLMs and stronger cloud LLMs to balance efficiency and performance (Narayan et al., 2025). A straightforward approach is to introduce a binary classifier as the router for weak-strong model selection (Ding et al., 2024; Ong et al., 2025). Beyond this, Chen et al. (2024) proposed routing queries based on the confidence of the small on-device model, while Aggarwal et al. (2024) introduced AutoMix, where the small model first generates and self-verifies its answer before a router decides whether to keep the local output or offload the query. Another line of work decomposes queries into subqueries and delegates them between device and cloud LLMs. Specifically, Division-of-Thoughts (Shao et al., 2025a) decomposes each query into dependent sub-tasks and assigns them to either a local model or a cloud LLM, while Route-and-Reason (Shao et al., 2025b) extends this idea to routing across multiple local and cloud models. Despite such advances, these works are either incompatible with our device-cloud collaborative reasoning setting or rely on external router, leaving the device model's intrinsic capacity for routing underutilized.

**Reinforcement learning for LLM post training.** RL-based post-training for LLM alignment was introduced in (Ziegler et al., 2019; Ouyang et al., 2022; Bai et al., 2022), where Proximal Policy Optimization (PPO) is the primary algorithm. However, due to PPO's complexity and computational cost, simplified alternatives have been proposed, including Direct Preference Optimization (Rafailov et al., 2023), ReMax (Li et al., 2024b), and Group Relative Policy Optimization (GRPO). Among these, GRPO (Shao et al., 2024) has gained particular traction for its simplicity and stability. Following the group sampling introduced in GRPO, several variants have been proposed that explore alternative optimization formulations, including DAPO, Dr. GRPO (Liu et al., 2025), CPPO (Lin et al., 2025), etc. While RL-based post-training has seen notable progress, a principled algorithmic framework for optimizing collaborative device-cloud LLM systems is still lacking.

**Incentivizing LLM reasoning via RL-based post-training.** The success of DeepSeek-R1 (Guo et al., 2025) has sparked sustained and growing interest in LLM reasoning, where the researchers proposed to apply RL and verifiable reward to cultivate the reasoning ability of base LLMs. Following this paradigm, Jin et al. (2025); Chen et al. (2025) proposed leveraging RL to improve LLM reasoning for better interaction with external search engines. Feng et al. (2025) further strengthen the long-horizon reasoning and tool-use abilities of LLMs via RL. More recently, RL has also been utilized to support LLM agents, where LLMs learn to plan and decompose tasks and call

for agentic tools (Wu et al., 2025). Nevertheless, incorporating LLM coordination within this paradigm and jointly optimizing reasoning and routing remain underexplored.

## 2. Problem Background

### 2.1. Device-Cloud LLM Collaboration

Consider a reasoning task with prompt set $\mathcal{D}$. The lightweight on-device LLM $\pi_\theta$, with tunable parameters $\theta$ designed for efficient deployment, may still struggle to handle certain prompts in $\mathcal{D}$ even after task-specific tuning, due to its limited capacity. To overcome this limitation, assistance from the cloud LLM $\pi_c$, which has a substantially broader knowledge scope, becomes essential. To maximize the potential of collaborative device-cloud LLMs on task $\mathcal{D}$, the most common solution is a two-stage pipeline: (i) tune the on-device model $\pi_\theta$ to enhance the local reasoning, and (ii) optimize a routing mechanism that decides whether each prompt should be handled locally or offloaded. We briefly review the most representative methods used for these two stages below.

**Stage I: Enhancing local reasoning via RL-based post-training.** A representative approach for this stage is GRPO (Shao et al., 2024). GRPO improves the on-device LLM by reinforcing relatively stronger responses while discouraging weaker ones among a group of candidates sampled from the model itself, enabling self-evolutionary learning. The detailed introduction to GRPO is presented in Appendix C. While such post-training strengthens the model's own reasoning ability for problem-solving, it does not endow the model with the capability to decide whether a prompt should be handled locally or offloaded, thereby necessitating a dedicated routing mechanism.

**Stage II: Routing optimization.** The existing approach is to train an additional binary classifier, often implemented as another LLM, to make routing decisions. For each prompt, a response is sampled from the on-device and cloud models, and the router is trained to distinguish whether the on-device LLM can solve the prompt or if it should be offloaded to the cloud LLM. Formally, given a dataset of prompts $\mathcal{D}$, we assign binary labels $z \in \{0, 1\}$ indicating whether the on-device model suffices for prompt $\boldsymbol{x}$. The router is then trained by minimizing the binary cross-entropy loss: $\mathcal{L}(\boldsymbol{w}) = -\frac{1}{|\mathcal{D}|} \sum_{\boldsymbol{x} \sim \mathcal{D}} (z \log p_w(\boldsymbol{x}) + (1-z) \log(1 - p_w(\boldsymbol{x})))$, where $p_w(\boldsymbol{x})$ denotes the router's predicted probability that the on-device LLM can handle $\boldsymbol{x}$ (Ding et al., 2024; Ong et al., 2025).

### 2.2. Limitations

The two-stage pipeline suffers from several inherent drawbacks. First, the router is essentially a binary classifier. It

is inherently difficult for such a classifier to judge whether the on-device LLM can solve a problem based solely on the surface-level prompt feature, particularly for tasks requiring complex reasoning, since problems with similar structures may vary greatly in difficulty. Conversely, using a more powerful LLM with reasoning ability as the router would be inefficient and wasteful, since making a routing decision would require duplicating the reasoning process that should be performed by the on-device LLM. This redundancy adds unnecessary computation and storage overhead without contributing to solving the task. Finally, training and maintaining the router introduce additional engineering overhead, adding complexity to the system.

## 3. RL-based Unified Training Methodology for Collaborative Reasoning

### 3.1. RL-based formulation

To address these limitations, we introduce a unified perspective that embeds routing optimization into post-training, allowing the on-device LLM to improve its reasoning ability for problem-solving while also learning routing strategies. Through fine-tuning parameters $\boldsymbol{\theta}$ of $\pi_\theta$, the on-device model not only strengthens its own problem-solving ability but also learns when to delegate to the cloud model $\pi_c$. Specifically, we anticipate the on-device LLM first attempts to generate a response locally and only invokes the cloud model $\pi_c$ at the end when it expects a better outcome. The resulting response $\boldsymbol{y}$ may be produced entirely by $\pi_\theta$ (i.e., $\boldsymbol{y} = \boldsymbol{y}^\theta$), or jointly with $\pi_c$ (i.e., $\boldsymbol{y} = [\boldsymbol{y}^\theta, \boldsymbol{y}^c]$).

To formalize this unified perspective, we cast training as a reward maximization problem, where the model seeks to optimize task performance subject to a budget on cloud model usage. This yields the following objective:

$$\max_{\boldsymbol{\theta}} \ \mathbb{E}_{\boldsymbol{x} \sim \mathcal{D}} \left[ R(\boldsymbol{\theta}, \boldsymbol{x}) \right] := \mathbb{E}_{\boldsymbol{x} \sim \mathcal{D}} \mathbb{E}_{\boldsymbol{y}^\theta \sim \pi_\theta(\boldsymbol{x})} [r(\boldsymbol{x}, \boldsymbol{y})]$$

subject to $\mathbb{E}[\mathbf{1}\{\boldsymbol{y} \sim (\pi_\theta, \pi_c)\}] \leq \rho \mathbb{E}[\mathbf{1}\{\boldsymbol{y} \sim \pi_\theta\}],$ (1)

where $\mathcal{D}$ is the prompt set, $r(\boldsymbol{x}, \boldsymbol{y})$ denotes the reward assigned to response $\boldsymbol{y}$ under prompt $\boldsymbol{x}$, and $\mathbb{E}[\mathbf{1}\{\boldsymbol{y} \sim (\pi_\theta, \pi_c)\}]$ and $\mathbb{E}[\mathbf{1}\{\boldsymbol{y} \sim \pi_\theta\}]$ represent the frequencies of invoking the cloud LLM and generating the response solely with the on-device LLM, respectively. We assume the cloud model $\pi_c$ generates deterministically without adding stochasticity to training (Shi et al., 2024). The constraint in Problem (1), governed by $\rho$, restricts the usage ratio between cloud and on-device LLMs, thereby limiting over-reliance on the cloud LLM and encouraging the on-device LLM to solve problems independently. Notably, we compute the reward over the entire response to reflect overall response quality, but update the on-device LLM $\pi_\theta$ using its generated portion.

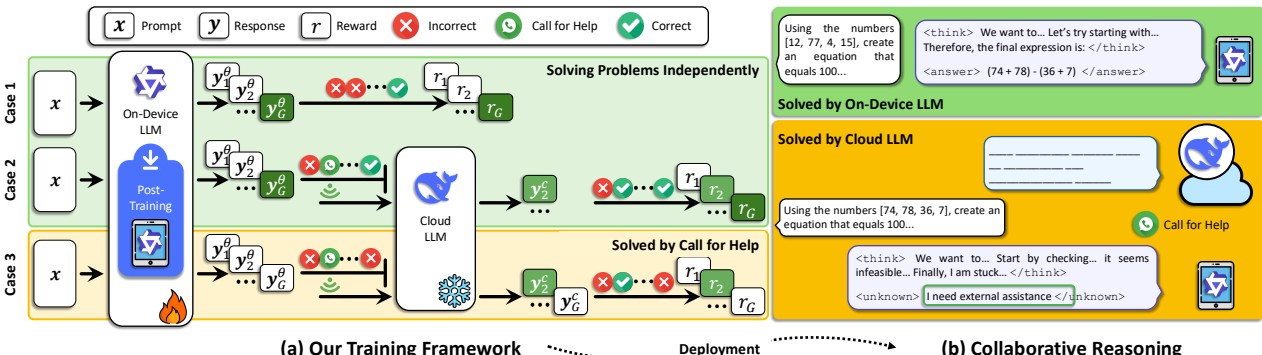

*Figure 1.* An illustration of our proposed RL-based unified training methodology and collaborative reasoning framework. (a) Training Framework: Two main scenarios where the lightweight on-device LLM learns to either solve problems independently or call for help. Note that the on-device LLM is trained offline before deployment on devices. (b) Collaborative Reasoning: The on-device LLM autonomously determines whether to process queries locally or invoke the cloud LLM.

## 3.2. Prompt and Hierarchical Rewards

The prompt and the reward function $r(\boldsymbol{x}, \boldsymbol{y})$ are two key components of Problem (1). In this subsection, we describe how we design the prompt template and reward to encourage the on-device model $\pi_\theta$ to invoke the cloud model $\pi_c$ when the task falls outside its capabilities.

**Prompt template.** As a next-token prediction model, an LLM tends to generate answers automatically, even when uncertain or incorrect. Without a dedicated prompt, it may fail to recognize its knowledge limitations and produce unreliable responses. To address this, we design a prompt template that guides the model to answer only when confident, and to invoke the cloud LLM for help when the question lies beyond its capabilities. The template is shown in Table 9 of Appendix D.

Such a template fosters the emergence of autonomous offloading behaviors. Consequently, the model could sample diverse strategies during initial training rollouts, either attempting the problem locally or requesting help, which creates a rich distribution of behaviors for the RL process. This allows us to design algorithm to reinforce successful offloading when the task exceeds the local model's capacity and encourage independent reasoning when tasks are within its scope, as illustrated in Figure 1.

**Hierarchical rewards.** Following the seminal work (Guo et al., 2025), we adopt a rule-based reward, which is well suited to reasoning tasks where verifiable rewards are typically available (Jin et al., 2025; Chen et al., 2025). To both foster effective coordination with the cloud LLM and maximize the on-device LLM's own problem-solving ability, we design a collaboration-aware hierarchical reward scheme comprising two components: *accuracy* and *coordination* rewards, as detailed below.

- *Accuracy reward:* This reward reflects the correctness of the response of the on-device model $\pi_\theta$. If the answer extracted from the on-device model $\pi_\theta$'s response is correct, an accuracy reward of $\alpha_a$ is assigned.
- *Coordination reward:* If the on-device model determines that it cannot solve the problem on its own and invokes the cloud LLM for assistance, a coordination reward of $\alpha_c$ is assigned, provided that the cloud LLM produces a correct answer.

We summarize all reward cases in (19) of Appendix D. Notably, these two rewards are in a competitive relationship and are mutually exclusive, i.e., a single response cannot receive both rewards simultaneously. In general, the reward weights satisfy $\alpha_a > \alpha_c$, aiming to prioritize the on-device model's independent problem-solving over its reliance on the cloud LLM. The relative weighting between accuracy reward and coordination reward will influence on-device LLM's propensity to offload. We treat it as a hyperparameter for balancing model's independent problem-solving with cloud invocation. The *theoretical intuition* behind hierarchical rewards is discussed in Appendix B.

## 3.3. Group-Adaptive Policy Gradient Algorithm

In this section, we propose a Group-Adaptive Policy Gradient algorithm, namely GAPG, tailored to Problem (1). The GAPG algorithm is characterized by *a group-level policy gradient*, designed to produce an unbiased variance reduced gradient estimator of the optimization objective, and *adaptive prompt filtering*, developed to ensure complementary learning signals for both independent problem solving and call for assistance. We introduced these two components in Sections 3.3.1 and 3.3.2, respectively, and summarize the overall procedures in Algorithm 1.

---

**Algorithm 1** Group-Adaptive Policy Gradient Algorithm for Collaborative Reasoning

---

**Require:** Initial on-device LLM $\pi_\theta$ with parameters $\boldsymbol{\theta}$, cloud LLM $\pi_c$, and prompt set $\mathcal{D}$

1: **for** iteration in $\{1, 2, \ldots, S\}$ **do**
2:     Sample a batch of prompts $\mathcal{D}_b$ from $\mathcal{D}$
3:     **for** each prompt $\boldsymbol{x} \in \mathcal{D}_b$ **do**
4:         Sample $G$ responses from the on-device LLM: $\{\boldsymbol{y}_1^\theta, \boldsymbol{y}_2^\theta, \ldots, \boldsymbol{y}_G^\theta\} \sim \pi_\theta(\cdot \mid \boldsymbol{x})$ and initialize $\boldsymbol{y}_i \leftarrow \boldsymbol{y}_i^\theta, \forall i$
5:         **if** any response in $\{\boldsymbol{y}_1^\theta, \boldsymbol{y}_2^\theta, \ldots, \boldsymbol{y}_G^\theta\}$ calls for help **then**
6:             Query cloud LLM $\pi_c$ to obtain $\boldsymbol{y}^c \sim \pi_c(\cdot \mid \boldsymbol{x})$         ▷ at most once for each prompt
7:             Set $\boldsymbol{y}_i \leftarrow [\boldsymbol{y}_i^\theta, \boldsymbol{y}^c]$ for each help-calling response $\boldsymbol{y}_i^\theta$ and set $\boldsymbol{y}_i \leftarrow \boldsymbol{y}_i^\theta$ for others    ▷ collaborative generation
8:         **end if**
9:         Evaluate rewards for responses $\{\boldsymbol{y}_1, \boldsymbol{y}_2, \ldots, \boldsymbol{y}_G\}$ based on (19)         ▷ hierarchical rewards
10:     **end for**
11:     Select prompts with both positive and negative responses, denoted as $\mathcal{D}_b^1$
12:     Select up to $\rho|\mathcal{D}_b^1|$ prompts for which none of the responses from $\pi_\theta$ are correct, but $\pi_c$ yields a correct answer, denoted as $\mathcal{D}_b^2$         ▷ adaptive prompt filtering
13:     Update $\boldsymbol{\theta} \leftarrow \boldsymbol{\theta} + \frac{\eta}{|\mathcal{D}_b^1 \cup \mathcal{D}_b^2|} \sum_{\boldsymbol{x} \in \mathcal{D}_b^1 \cup \mathcal{D}_b^2} \widehat{\nabla_\theta R}(\boldsymbol{\theta}, \boldsymbol{x})$         ▷ group-adaptive policy gradient
14: **end for**
15: **return** the updated on-device model $\pi_\theta$

---

### 3.3.1. GROUP-LEVEL POLICY GRADIENT

The reward function $r(\boldsymbol{x}, \boldsymbol{y})$ is non-differentiable with respect to the model parameters $\boldsymbol{\theta}$ as it does not admit an analytic expression in terms of $\boldsymbol{\theta}$. Consequently, standard gradient-based optimization algorithms cannot be directly applied. To address this challenge, we turn to policy gradient (Sutton et al., 1999) and consider the expected reward defined in (1) which is differentiable to the policy model. Inspired by the group sampling strategy in GRPO, we introduce a group-level gradient estimator, aiming to reduce the gradient variance and stabilize the overall training. Proposition 3.1 states the formulation and key properties of the group-level policy gradient estimator. The proof can be found in Appendix F.

**Proposition 3.1** (Group-level Policy Gradient Estimator). Given a prompt $\boldsymbol{x}$, draw a group of $G$ responses $\{\boldsymbol{y}_1, \ldots, \boldsymbol{y}_G\}$, where each response $\boldsymbol{y}_i$ may be produced entirely by the on-device policy $\pi_\theta$ (i.e. $\boldsymbol{y}_i = \boldsymbol{y}_i^\theta$) or jointly with the cloud policy $\pi_c$ (i.e. $\boldsymbol{y}_i = [\boldsymbol{y}_i^\theta, \boldsymbol{y}^c]$). Denote the reward for response $i$ as $r_i = r(\boldsymbol{x}, \boldsymbol{y}_i)$ and the group mean reward $\bar{r} = \frac{1}{G} \sum_{i=1}^G r_i$.

- For any $G \geq 2$, the following quantity

$$\widehat{\nabla_\theta R}(\boldsymbol{\theta}, \boldsymbol{x}) = \frac{1}{G-1} \sum_{i=1}^G \nabla_\theta \log \pi_\theta(\boldsymbol{y}_i^\theta \mid \boldsymbol{x})(r_i - \bar{r}) \quad (2)$$

is an *unbiased* estimator of the policy gradient $\nabla_\theta R(\boldsymbol{\theta}, \boldsymbol{x}) = \nabla_\theta \mathbb{E}_{\boldsymbol{y}^\theta \sim \pi_\theta(\boldsymbol{x})}[r(\boldsymbol{x}, \boldsymbol{y})]$.
- The covariance and variance of the group-level policy gradient estimator (2) scale as $\mathcal{O}(1/G)$ with the group

size, i.e., $\text{Cov}\left(\widehat{\nabla_\theta R}(\boldsymbol{\theta}, \boldsymbol{x})\right) \sim \mathcal{O}(\frac{1}{G})$ and

$$\mathbb{E}\left[\left\|\widehat{\nabla_\theta R}(\boldsymbol{\theta}, \boldsymbol{x}) - \nabla_\theta R(\boldsymbol{\theta}, \boldsymbol{x})\right\|_2^2\right] \sim \mathcal{O}(\frac{1}{G}). \quad (3)$$

By leveraging the group averaging effect (i.e., $G$) and a sample-based baseline (i.e., $\bar{r}$), the group-level policy gradient estimator in (2) enjoys a lower variance. With this gradient estimator, a straightforward approach is to iteratively update the parameter of the on-device LLM $\pi_\theta$ via stochastic gradient ascent:

$$\boldsymbol{\theta} \leftarrow \boldsymbol{\theta} + \eta \widehat{\nabla_\theta R}(\boldsymbol{\theta}, \boldsymbol{x}), \quad \boldsymbol{x} \sim \mathcal{D}, \quad (4)$$

where $\eta$ is the learning rate. However, the update in (4) doesn't account for the constraint on frequency of invoking the cloud LLM in Problem (1). If the sampled prompts predominantly trigger external assistance in the early stages of training, the model may suffer from a scarcity of informative gradient signals required to enhance its local reasoning capabilities. Without intervention, this imbalance can lead the policy to degenerate into a state that relies exclusively on cloud invocation, thereby stunting the development of independent problem-solving ability.

### 3.3.2. GROUP-ADAPTIVE PROMPT FILTERING

To further encourage the on-device LLM $\pi_\theta$ to explore and limit the frequency of invoking the cloud LLM $\pi_c$, we introduce an adaptive prompt filtering mechanism, guided by the constraint in Problem (1). Unlike prior methods that focus primarily on improving standalone problem-solving ability, our filtering strategy is designed to explicitly support collaboration-aware learning.

The key principle is that, during training, the on-device LLM should be exposed to both prompts it can solve independently and prompts that exceed its capability. Training exclusively on locally solvable prompts biases the model toward local reasoning, while focusing only on prompts that require external assistance leads to over-reliance on the cloud. By jointly observing both cases, the model learns to distinguish task difficulty and develop appropriate routing behavior between independent problem solving and assistance seeking. Specifically, in each round, we sample $G$ responses for each prompt $x \in \mathcal{D}_b$ using models $[\pi_\theta, \pi_c]$. Based on the responses, we form two prompt subsets:

- Set $\mathcal{D}_b^1$: Includes prompts where at least one of the $G$ sampled responses is generated correctly by the on-device model $\pi_\theta$. These prompts help the model learn to solve tasks on its own.
- Set $\mathcal{D}_b^2$: Includes up to $\rho|\mathcal{D}_b^1|$ prompts for which none of the sampled responses from on-device model $\pi_\theta$ are correct, but the cloud LLM $\pi_c$ provides a correct answer. These prompts are essential for guiding the model to identify situations in which leveraging the cloud LLM is beneficial. The filtering ratio inherits from the constraint in Problem (1).

Accordingly, the iterative parameter update for the on-device model can be expressed as:

$$\boldsymbol{\theta} \leftarrow \boldsymbol{\theta} + \frac{\eta}{|\mathcal{D}_b^1 \cup \mathcal{D}_b^2|} \sum_{\boldsymbol{x} \in \mathcal{D}_b^1 \cup \mathcal{D}_b^2} \widehat{\nabla_\theta R}(\boldsymbol{\theta}, \boldsymbol{x}). \quad (5)$$

By training $\pi_\theta$ on $\mathcal{D}_b^1 \cup \mathcal{D}_b^2$, the on-device LLM receives complementary learning signals for both independent problem solving and calling for assistance. Overall, prompt filtering and hierarchical rewards collaboratively empower the on-device model to balance independent problem-solving with calling for assistance. By tuning these components, we enable the model to maximize its potential while adhering to offloading constraints. The theoretical justification of our methodology is provided in Appendix B.

**Remark 3.2** (Reliance on pass@$G$). Our prompt filtering does not rely solely on the current pass@1 accuracy of the on-device model. For each prompt, we sample $G$ responses from the on-device model, and the prompt is included in $\mathcal{D}_b^1$ as long as at least one sampled response is correct. Therefore, even when pass@1 is relatively low, $\mathcal{D}_b^1$ is unlikely to be empty if the model has a sufficiently strong pass@$G$. In practice, if $\mathcal{D}_b^1$ becomes too small, one can increase $G$, which raises the probability of obtaining at least one correct on-device response and thus enlarges $\mathcal{D}_b^1$. If the model is extremely weak and pass@$G$ remains close to zero even for a large $G$, then $D_b^1$ may indeed become empty. However, this is not a failure mode specific to our method; rather, it reflects a broader limitation of on-policy RL post-training methods.

**Remark 3.3** (Comparison between GAPG and GRPO). Although the proposed GAPG algorithm adopts the same group sampling mechanism as GRPO, it is fundamentally different in both its objective and formulation. GRPO and its variants (e.g., Dr. GRPO (Liu et al., 2025)) are primarily designed to enhance the reasoning capability of LLMs within a conventional fine-tuning paradigm. In contrast, GAPG targets a device-cloud collaborative setting, where the key challenge lies not only in improving on-device reasoning performance but also in jointly optimizing local reasoning and cloud invocation strategy under budget constraints. A more detailed technical comparison between GAPG and GRPO (including its variant Dr. GRPO) is provided in Appendix C.

## 4. Experiments

**Datasets.** We train and evaluate on-device LLMs using Countdown (Pan et al., 2025) and MATH-lighteval (Hendrycks et al., 2021). To test generalization, models fine-tuned on MATH-lighteval are further benchmarked against MATH-500 (Hendrycks et al., 2021), AMC23 (Lewkowycz et al., 2022), MinervaMath (Li et al., 2024a), and AGI-Eval-Math (Zhong et al., 2024).

**On-device and cloud LLMs.** We employ Deepseek-R1 as the cloud model. For the Countdown task, we adopt Qwen2.5-3B-Instruct as the on-device model. For the MATH-lighteval task, we evaluate three on-device models of different sizes, Llama-3.2-1B-Instruct, Qwen2.5-1.5B-Instruct, and Llama-3.2-3B-Instruct. For all device-cloud collaboration scenarios in Section 4.1.1, the call-for-cloud ratio is constrained to $30\%$ (i.e., $\rho/(1 + \rho)$), with requests exceeding this threshold redirected to the on-device LLM. Section 4.1.2 builds on this setting and further studies the impact of varying the ratio on performance.

**Baselines.** The following methods are used for comparison.

- Cloud LLM: All the queries are offloaded to the cloud model, i.e, Deepseek-R1 (Guo et al., 2025), which serves as a performance upper bound.
- Task-Tuning Only: Perform task-specific fine-tuning on the on-device model using GRPO (Shao et al., 2024). During inference, all queries are processed locally.
- Task-Tuning&Naive Offloading: The on-device LLM is first fine-tuned as in Task-Tuning Only. During inference, a certain proportion of queries are randomly offloaded to the cloud LLM.
- Task-Tuning&Router: The on-device LLM is first fine-tuned as in Task-Tuning Only, and then a router (DeBERTa-v3-large) is trained to decide whether to call the cloud LLM based on the query (Ding et al., 2024).
- Collaboration-Aware Tuning: The on-device LLM is fine-tuned via Dr. GRPO (Liu et al., 2025), augmented by

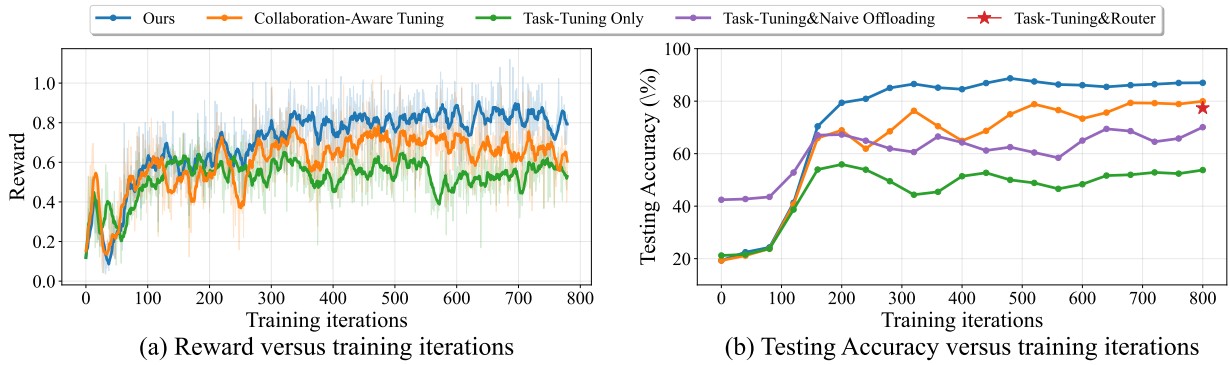

(a) Reward versus training iterations

(b) Testing Accuracy versus training iterations

*Figure 2.* Training reward and testing accuracy on the Countdown task with Qwen2.5-3B-Instruct. Our method consistently outperforms baselines, achieving higher rewards and accuracy.

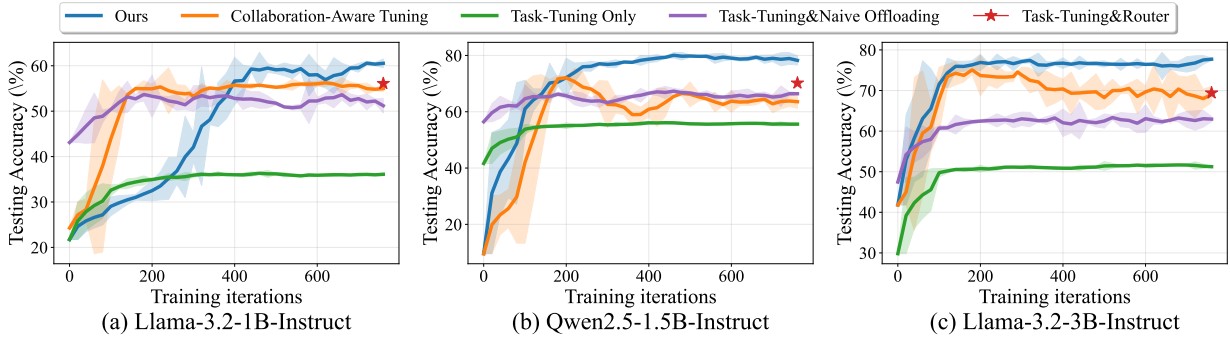

(a) Llama-3.2-1B-Instruct

(b) Qwen2.5-1.5B-Instruct

(c) Llama-3.2-3B-Instruct

*Figure 3.* Testing accuracy versus training iterations on the MATH-lighteval dataset. Our method consistently outperforms baselines across three on-device models, while also exhibiting stable training behavior, demonstrating its effectiveness and robustness.

our proposed hierarchical rewards (i.e., (19)) to encourage collaboration. The exploration on combining hierarchical rewards with GRPO is provided in Appendix C.

For the Task-Tuning&Router baseline, we adopt a lightweight classifier to ensure compatibility with on-device deployment. In addition, we evaluate a reasoning-based router, Router-R1 (Zhang et al., 2026), in Appendix A.6.

### 4.1. Main Results

#### 4.1.1. SYMBOLIC REASONING: COUNTDOWN TASK

Figure 2 compares the training reward and accuracy of our approach and the baselines on the Countdown task using the Qwen2.5-3B-Instruct model. Since Task-Tuning&Naive Offloading and Task-Tuning&Router employ the same RL process as Task-Tuning Only for tuning the on-device LLM, we report only the reward of Task-Tuning Only in Figure 2(a). As shown, our approach steadily achieves higher rewards and converges to a superior plateau compared to the baselines, indicating more efficient learning and stronger device-cloud coordination. Collaboration-Aware Tuning with our proposed hierarchical rewards outperforms Task-Tuning Only, demonstrating

the effectiveness of the hierarchical reward design and the proposed collaboration scheme. However, it still underperforms our method, highlighting the advantages of our proposed algorithm. This improvement can be attributed to the fact that our algorithm is more closely aligned with the overall problem formulation, as discussed in Appendix C.

Figure 2(b) compares the testing accuracy of our method against baselines. Since the router is trained only after the completion of task-specific training, we report only the final accuracy of Task-Tuning&Router in Figure 2(b) and the training process of the router is provided in Appendix A.5. As shown in this figure, our method approaches Cloud LLM performance and surpasses all baselines. In particular, it improves accuracy by approximately 30%, nearly matching the cloud offloading ratio, relative to Task-Tuning Only. This demonstrates that our approach equips the on-device model with coordination capability without compromising its problem-solving ability, highlighting the superiority of our collaborative unified training methodology.

#### 4.1.2. MATHEMATICAL REASONING

**Training dynamics.** We further evaluate our approach on the MATH-lighteval dataset using three on-device models of varying sizes (1B, 1.5B, and 3B). The testing ac-

*Table 1.* Testing accuracy (%) of our approach and baselines with Qwen2.5-1.5B-Instruct and Llama-3.2-3B-Instruct, tuned on MATH-lighteval. Conditional local accuracy (Cond. Acc.) is defined as the ratio of correct outputs produced by the on-device LLM to the total number of problems retained after offloading. Our method achieves the highest average accuracy across both models.

| Model | Metric | Method | MATH-lighteval | MATH-500 | AMC23 | MinervaMath | AGI-Eval-Math | Avg. |
|---|---|---|---|---|---|---|---|---|
| | | Cloud LLM | 98.4 | 97.3 | 97.5 | 80.9 | 94.7 | 93.8 |
| Qwen2.5-1.5B | Cond. Acc. | Task-Tuning Only | 56.1 | 54.8 | 35.0 | 20.6 | 51.8 | 43.7 |
| | | Task-Tuning&Naive Offloading | 56.3 | 55.4 | 32.1 | 21.0 | 51.6 | 43.3 |
| | | Collaboration-Aware Tuning | 54.1 | 61.2 | 28.6 | 13.6 | 55.3 | 42.6 |
| | | Task-Tuning&Router | 61.8 | 64.9 | 39.3 | 21.8 | 58.7 | 49.3 |
| | | **Ours** | **72.6** | **75.1** | **42.9** | **24.1** | **64.6** | **55.8** |
| | Overall Acc. | Task-Tuning Only | 56.1 | 54.8 | 35.0 | 20.6 | 51.8 | 43.7 |
| | | Task-Tuning&Naive Offloading | 67.2 | 67.4 | 50.0 | 38.2 | 68.3 | 58.2 |
| | | Collaboration-Aware Tuning | 61.5 | 61.2 | 42.5 | 33.5 | 66.9 | 53.1 |
| | | Task-Tuning&Router | 70.9 | 72.2 | 55.0 | 36.8 | 69.2 | 60.8 |
| | | **Ours** | **80.4** | **81.6** | **57.5** | **40.8** | **73.4** | **66.7** |
| Llama-3.2-3B | Cond. Acc. | Task-Tuning Only | 51.2 | 43.0 | 27.5 | 19.1 | 45.5 | 37.3 |
| | | Task-Tuning&Naive Offloading | 51.6 | 41.8 | 27.5 | 19.4 | 45.7 | 37.2 |
| | | Collaboration-Aware Tuning | 60.2 | 43.7 | 21.4 | 13.4 | 43.9 | 36.5 |
| | | Task-Tuning&Router | 64.9 | 45.7 | 25.0 | 20.2 | 35.1 | 38.2 |
| | | **Ours** | **72.2** | **56.6** | **35.7** | **27.7** | **59.7** | **50.4** |
| | Overall Acc. | Task-Tuning Only | 51.2 | 43.0 | 27.5 | 19.1 | 45.5 | 37.3 |
| | | Task-Tuning&Naive Offloading | 65.1 | 59.0 | 45.0 | 37.1 | 58.7 | 53.0 |
| | | Collaboration-Aware Tuning | 66.8 | 59.6 | 42.5 | 36.8 | 58.9 | 52.9 |
| | | Task-Tuning&Router | 69.4 | 62.0 | 46.0 | 39.1 | 53.8 | 54.1 |
| | | **Ours** | **79.5** | **68.6** | **52.5** | **43.4** | **64.5** | **61.7** |

curacy over training iterations is presented in Figure 3. While our method lags behind some baselines in the early training phase, it consistently surpasses them as training progresses and ultimately achieves the highest accuracy across all model sizes. In contrast, the Task-Tuning&Naive Offloading scheme shows a competitive performance at the early stage, particularly with the Qwen-1.5B-Instruct model. This is primarily because this scheme employs a fixed cloud-invocation strategy that is independent of the training process. Additionally, the slower early-stage convergence of our approach arises because our methodology explicitly balances on-device LLM's own problem-solving ability with cloud LLM coordination. Nevertheless, as Figure 3 demonstrates, this joint optimization yields clear long-term benefits: our approach converges to substantially higher accuracy, underscoring its advantages.

**Evaluation outside the training task.** Beyond MATH-lighteval, we report the performance of the tuned Qwen2.5-1.5B and Llama-3.2-3B models on four additional widely used mathematical benchmarks in Table 1. The conditional local accuracy measures the correctness of the on-device LLM on problems it attempts to solve independently after offloading. As shown in Table 1, our method achieves the highest overall and conditional local accuracy across all benchmarks and both models. In particular, the improved conditional local accuracy indicates that our offloading strategy effectively filters out difficult queries, allowing the on-device LLM to focus on high-confidence problems it can solve reliably. Additionally, both the problem-solving and routing strategy generalize well to unseen tasks, indicating that reasoning abilities learned during training are

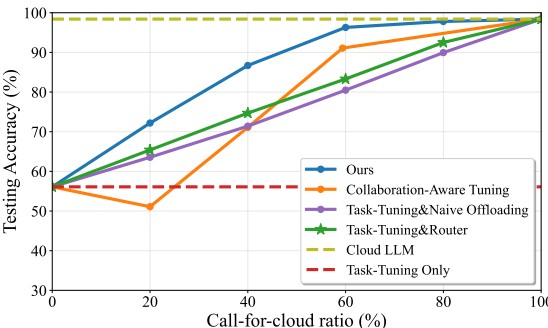

*Figure 4.* Performance under varying cloud usage. Our approach rapidly narrows the gap to Cloud LLM as the ratio increases.

inherently more transferable across tasks than the external binary router, leading to more robust coordination.

**Impact of call-for-cloud ratio.** In Figure 4, we evaluate the impact of the call-for-cloud ratio on the testing accuracy. For the Collaboration-Aware Tuning baseline, the on-device model converges to a call-for-cloud ratio of approximately 59.8%. To evaluate it at lower target ratios (e.g., 20% or 40%), we truncate additional calls once the target ratio is reached. As shown in Figure 4, our method achieves the strongest performance across all ratios, delivering notable gains even with moderate cloud reliance (20-40%) and nearly matching the Cloud LLM at 60%. In contrast, Collaboration-Aware Tuning suffers performance degradation at low ratios (e.g., 20%), as it fails to effectively balance the development of coordination with strengthening problem-solving. Meanwhile, the performance of Task-Tuning&Naive Offloading and Task-Tuning&Router improves steadily but consistently remains inferior. Overall,

the proposed collaborative unified training consistently out-performs separate routing and naive offloading.

## 4.2. Further Experiments

We conduct ablation studies to examine the impact of the policy gradient group size, hierarchical rewards, and prompt filtering on performance. Full results are provided in Appendix A.1. We compare costs in terms of total cloud invocations during training in Appendix A.2 and evaluate the token costs of independent reasoning and calling for assistance in Appendix A.3, which together validate the efficiency of our proposed methodology. We provide an analysis of call-for-help traces in Appendix A.4 to better understand the learned assistance-seeking behavior. We further study the limitation of classifier-based router in Appendix A.5. Additionally, we consider more baselines in Appendix A.6 and evaluate the robustness of our method when switching the cloud LLM at inference time in Appendix A.7.

## 5. Conclusion

We proposed a collaborative device-cloud LLM reasoning framework where the on-device LLM itself decides whether to invoke the cloud LLM at the inference time. To endow this capability, we formulated a hierarchical reward maximization problem that integrates routing optimization into post-training, enabling the on-device LLM to strengthen its problem-solving ability while developing effective coordination with the cloud LLM. To solve this problem, we developed a group-adaptive policy gradient algorithm featuring a group-level policy gradient for provably stable optimization, alongside adaptive prompt filtering to provide complementary learning signals for both local problem solving and assistance invocation. Experiments across diverse models and benchmarks confirmed the effectiveness of our proposed methodology.

## Limitations

One limitation of our work is that we focus on reasoning tasks with readily available correctness-based rewards. Extending our framework to more general collaborative inference settings, such as open-ended tasks, would require the design of reliable quality-aware reward functions (e.g., LLM-as-a-Judge). Advancing this aspect represents an important direction for future research.

## Impact Statement

This paper contributes a unified framework for on-device and cloud LLM collaboration that integrates cloud coordination optimization directly into the reinforcement learning

(RL)-based post-training of the on-device model, eliminating the need for separate routing mechanisms. The focus of this work is on the technical advancement of device-cloud LLM collaboration. While this research has potential societal impacts, it primarily addresses technical challenges and does not necessitate a specific discussion on societal consequences.

## Acknowledgments

This work was supported in part by the National Science Foundation (NSF) under Grants CNS-2146171 and CPS-2313109; by the Office of Naval Research (ONR) under Grant N00014-21-1-2472; by the Air Force Office of Scientific Research (AFOSR) under Grant FA9550-24-1-0083; and by the National Research Foundation of Korea (NRF) grant funded by the Korea government (MSIT) (No. RS-2026-25470983).

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

# Appendix

# A. Further Experiments

## A.1. Ablation Studies

To validate the contribution of each component in our framework, we conduct ablation studies on the policy gradient group size, hierarchical rewards, and prompt filtering.

### A.1.1. IMPACT OF THE GROUP SIZE OF THE POLICY GRADIENT

The group-level policy gradient (2) serves as an unbiased estimator of the true gradient. As established in Proposition 3.1, the gradient variance scales inversely with the group size. In this section, we apply the proposed algorithm to fine-tune Qwen2.5-1.5B-Instruct and Llama-3.2-3B-Instruct on the MATH-lighteval dataset. We evaluate the training dynamics of the proposed algorithm across various group sizes, maintaining all the other components identical, to observe their impact on performance.

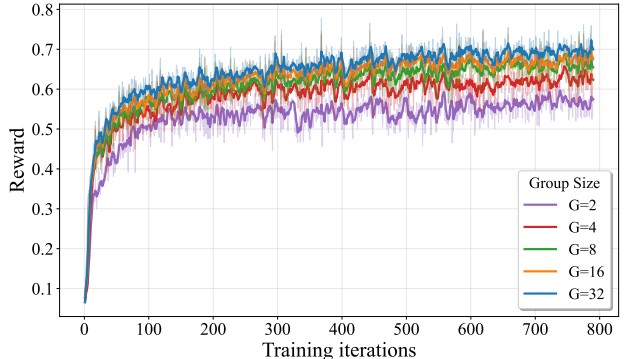
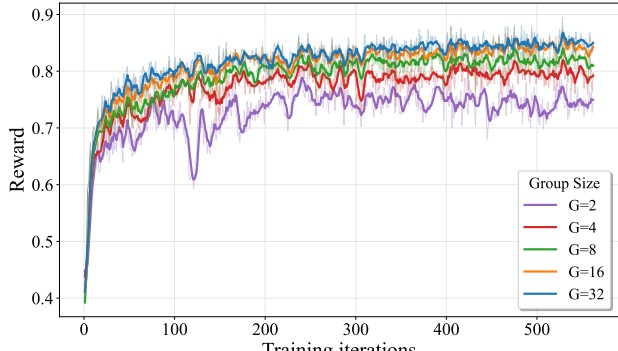

*Figure 5.* Impact of the group size $G$ on the convergence of the proposed algorithm over Qwen2.5-1.5B-Instruct model and MATH-lighteval dataset.

*Figure 6.* Impact of the group size $G$ on the convergence of the proposed algorithm over Llama-3.2-3B-Instruct model and MATH-lighteval dataset.

*Table 2.* Impact of the group size $G$ on the testing accuracy (%) of the proposed algorithm over the MATH-lighteval dataset.

| Model / Group size | $G = 2$ | $G = 4$ | $G = 8$ | $G = 16$ | $G = 32$ |
|---|---|---|---|---|---|
| Qwen2.5-1.5B-Instruct | 79.1 | 79.9 | 80.4 | 80.6 | 80.6 |
| Llama-3.2-3B-Instruct | 77.5 | 79.1 | 79.5 | 79.8 | 79.9 |

Figures 5 and 6 present the convergence behavior of the proposed algorithm when tuning Qwen2.5-1.5B-Instruct and Llama-3.2-3B-Instruct with different group sizes, respectively. Consistent with Proposition 3.1, increasing the group size reduces gradient variance, leading to more stable optimization and higher final rewards. Smaller group sizes exhibit noisier learning curves and occasional performance drops, whereas larger group sizes produce smoother and more reliable convergence. Although performance improves as the group size increases, the gains saturate beyond $G > 16$, indicating diminishing returns at very large group sizes. On the other hand, larger group sizes introduce additional computational overhead due to increased sampling and evaluation costs. Considering both the diminishing performance gains and the rising computational burden, a moderate group size provides a favorable trade-off between efficiency and performance. These trends are further reflected in the testing accuracies reported in Table 2.

### A.1.2. IMPACT OF THE HIERARCHICAL REWARDS

The hierarchical reward is a core component of our methodology. As discussed in Section 3.2, we assign accuracy reward $\alpha_a$ when the on-device LLM solves a problem independently and a coordination reward $\alpha_c$ when it successfully invokes the cloud LLM to obtain the correct answer. This reward structure incentivizes the model to strength both autonomous problem-solving and strategic offloading. However, the pattern for invoking the cloud LLM is generally easier for the model to learn than diverse independent reasoning patterns. Consequently, the relative magnitude of accuracy reward $\alpha_a$ versus coordination reward $\alpha_c$ is thus critical for ensuring that the on-device LLM maximizes its potential in both local execution and cloud coordination.

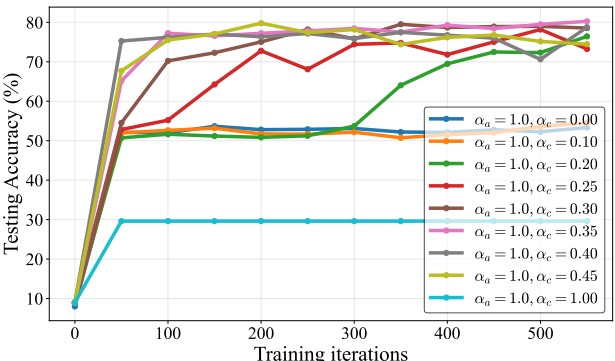 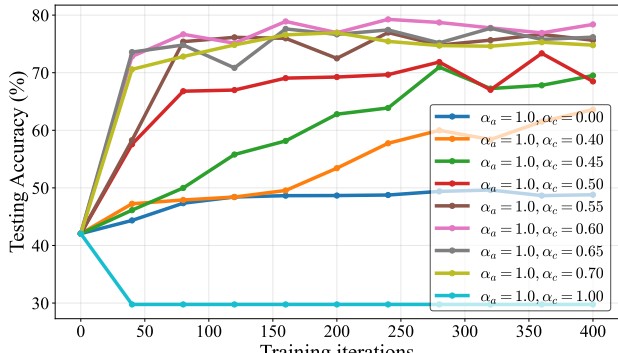

*Figure 7.* Impact of the hierarchical reward weights on the performance of the proposed algorithm over Qwen2.5-1.5B-Instruct model and MATH-lighteval dataset. $\alpha_a$ denotes the accuracy reward and $\alpha_c$ represents the coordination reward.

*Figure 8.* Impact of the hierarchical reward weights on the performance of the proposed algorithm over Llama-3.2-3B-Instruct model and MATH-lighteval dataset. $\alpha_a$ denotes the accuracy reward and $\alpha_c$ represents the coordination reward.

Note that the attained reward during training is sensitive to the choice of reward weights. We thus evaluate performance based on testing accuracy rather than the raw attained reward as in Appendix A.1.1. In Figures 7 and 8, we report the testing accuracies of Qwen2.5-1.5B-Instruct and Llama-3.2-3B-Instruct across training iterations under different combinations of $\alpha_a$ and $\alpha_c$. We fix the accuracy reward at $\alpha_a = 1.0$ and vary the coordination reward $\alpha_c$ over a specific range. When the coordination reward is set to $\alpha_c = 0$, the achieved accuracy is relatively low, as this setting does not encourage offloading to the cloud LLM. As $\alpha_c$ increases, both models achieve higher accuracy by leveraging assistance from the cloud model. However, when $\alpha_c$ further increases to $1.00$, the on-device LLM loses its reasoning capability and collapses into a policy that always requests help. Since the call-for-help ratio is capped at $30\%$, the overall accuracy becomes trapped at around $30\%$. As shown in Figures 7 and 8, when the hierarchical reward weights are set within an appropriate range, our proposed algorithm effectively enhances both problem-solving performance and cloud coordination. These results demonstrate the importance of the proposed hierarchical rewards.

**Remark A.1** (Practical Strategy for Setting $\alpha_c$.)**.** Strong performance is achieved over a reasonably broad interval rather than at a single sharp optimum. Concretely, in Figure 7 (Qwen2.5-1.5B), strong performance is maintained for roughly $\alpha_c \in [0.25, 0.45]$, while in Figure 8 (Llama-3.2-3B), the stable high-performing region is around $\alpha_c \in [0.55, 0.70]$. Thus, the effective region spans multiple neighboring values in each case. Practically, we handle this by using a small set of widely spaced $\alpha_c$ candidates (e.g., $\{0.2, 0.4, 0.6, 0.8\}$) rather than an exhaustive task-specific grid search. Inappropriate values are easy to identify because they typically cause rapid early collapse toward one-sided coordination behavior, which becomes visible early in training. Therefore, for a new model, a coarse search over $(0, 1)$ with early monitoring is a practical tuning strategy.

### A.1.3. IMPACT OF THE PROMPT FILTERING

Prompt filtering aims to ensure that the on-device model receives complementary learning signals for both independent problem solving and requesting assistance. This balance is crucial for strengthening both behaviors during training. For a pretrained initial model, the response behavior during rollout is typically uncontrollable, and the ratio between self-solving and requesting assistance is unclear. If either behavior dominates, it may suppress the other and bias the model toward a suboptimal policy of either exclusive local execution or exclusive cloud offloading.

We evaluate Qwen2.5-1.5B-Instruct, Qwen2.5-3B-Instruct, and Llama-3.2-3B-Instruct on the Countdown and MATH-lighteval datasets, with the prompt filtering ratio set to $30\%$. During testing, any requests exceeding this budget are automatically intercepted and processed on-device. As shown in Figure 9, prompt filtering leads to a steady increase in testing accuracy across all tasks, stabilizing the overall training. In contrast, removing prompt filtering results in a noticeable performance drop. A possible explanation is that, without prompt filtering, the on-device model fails to adhere to the cloud LLM invocation budget during training, leading to an imbalance between local execution and cloud offloading. When a hard offloading threshold is applied at test time to truncate excess requests, this mismatch causes degraded performance. This effect is particularly pronounced for the Llama-3.2-3B-Instruct model, where we observe significant performance oscillations. These results validate the importance of prompt filtering for maintaining stable and effective device-cloud

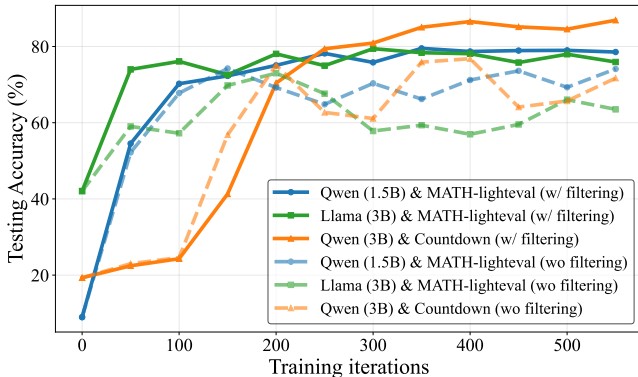

*Figure 9.* Impact of the prompt filtering on the performance of the proposed algorithm over Qwen2.5-1.5B-Instruct, Qwen2.5-3B-Instruct, and Llama-3.2-3B-Instruct models and Countdown and MATH-lighteval datasets.

collaboration.

*Table 3.* Impact of the prompt filtering ratio on call-for-help behavior and accuracy (%) based on Qwen2.5-1.5B-Instruct model and MATH-lighteval dataset. Conditional local accuracy is defined as the ratio of correct outputs produced by the on-device LLM to the total number of problems retained after offloading. During evaluation, the effective call-for-help ratio used to compute test accuracy is capped by the corresponding cloud usage budget.

| Filtering ratio (cloud budget) | Local accuracy | Conditional local accuracy | Call-for-help ratio | Overall accuracy |
|---|---|---|---|---|
| 20 | 55.1 | 68.9 | 21.3 | 74.3 |
| 30 | 52.6 | 72.2 | 28.6 | 80.4 |
| 40 | 49.7 | 89.3 | 44.2 | 88.2 |
| 50 | 41.3 | 90.1 | 53.7 | 90.7 |

In Table 3, we investigate the model's behavior under different prompt filtering ratios, which correspond to varying cloud offloading budgets. The conditional local accuracy is defined as the ratio of correct local outputs to the total number of problems the local model attempts to solve independently. As the prompt filtering ratio increases, the model more frequently delegates queries to the cloud model, leading to a higher call-for-help ratio and a lower local accuracy. This trend is expected, as the local model increasingly relies on external assistance for challenging queries. Meanwhile, the conditional local accuracy consistently improves with higher filtering ratios. This indicates that the problems retained for local processing are predominantly high-confidence cases that the on-device model can solve reliably. Importantly, despite the decline in raw local accuracy, the overall system accuracy improves as the filtering ratio increases. This demonstrates that strategic cloud offloading enables the system to leverage the stronger cloud model for difficult queries, thereby achieving better overall performance. Moreover, these results highlight that the prompt filtering ratio is a critical control variable for balancing the trade-off between local processing, cloud reliance, and overall system accuracy.

### A.2. Training Cost in Cloud LLM Usage

Among the considered baselines, all methods rely on RL-based fine-tuning and thus share similar policy update complexity. Our approach introduces additional cost due to cloud LLM calls. Notably, Collaboration-Aware Tuning and Task-Tuning&Router also depend on the cloud LLM during training. As specified in Step 6 of Algorithm 1, the cloud LLM is invoked at most once per prompt in our method. We cache the cloud response upon the first invocation and reuse it across subsequent training iterations. Consequently, the total cloud-call cost is bounded by the training set size. Task-Tuning&Router queries the cloud LLM only when the on-device model fails, and this result is also stored for router training. We report the number of total cloud invocations across these approaches in Table 4, where our method exhibits cloud invocation costs comparable to these baselines. Overall, the response caching and reuse save the cloud invocation costs, making the additional overhead controlled and comparable across these approaches.

*Table 4.* Total number of cloud LLM invocations during the training process. These results are obtained using the same hyperparameter configurations as those in Figures 2 and 3.

| Method | Countdown Task | | MATH-lighteval Task | | Avg. |
| --- | --- | --- | --- | --- | --- |
| | **Qwen2.5-3B-Instruct** | **Llama-3.2-1B-Instruct** | **Qwen2.5-1.5B-Instruct** | **Llama-3.2-3B-Instruct** | |
| **Task-Tuning&Router** | 2238 | 3845 | 2540 | 3075 | 2925 |
| **Collaboration-Aware Tuning** | 3266 | 3920 | 3184 | 2237 | 3152 |
| **Ours** | 2451 | 3796 | 2519 | 2883 | 2913 |

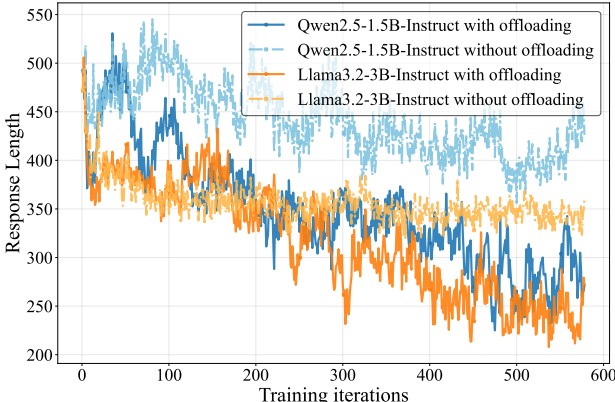

*Figure 10.* Comparison of response length of on-device LLM rollouts between with (ours) and without cloud offloading over the Qwen2.5-1.5B-Instruct and Llama-3.2-3B-Instruct models on the MATH-lighteval dataset during training.

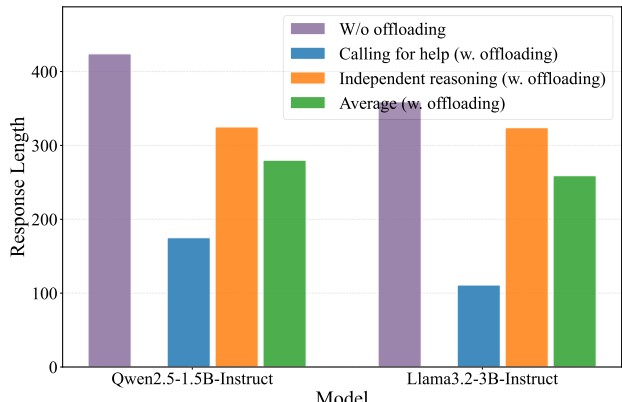

*Figure 11.* Detailed comparison of response length across different categories at test time. W/o offloading represents the setting where the on-device LLM attempts to solve all the problems independently. The other three categories denotes different cases in our device-cloud collaboration scenario.

### A.3. Token Efficiency of On-Device LLM

In Figures 10, we compare the response length of the on-device LLM with and without cloud offloading. In the former setting, the on-device LLM either solves problems independently or calls for assistance from the cloud LLM. In the latter setting, the on-device LLM must solve all problems independently. As shown in the figure, the response length with cloud offloading is lower than that without offloading. One potential reason is that difficult problems are offloaded to the cloud, allowing the on-device model to focus only on easier problems. Moreover, as training progresses, the response length decreases, which can be attributed to the improved routing ability of the model, enabling it to better distinguish between difficult and easy problems.

In Figure 11, we provide a detailed comparison of response length at test time, reporting the response length for different categories under the offloading setting, including calling for help, independent reasoning, and their average. As shown, the on-device LLM uses fewer tokens to solve the remaining problems after offloading compared with solving all problems independently (i.e., the purple bar). In addition, the response length for calling for help is shorter, which implies that the on-device model makes routing decisions efficiently, incurring little token cost to recognize when a problem is beyond its capacity. Overall, the average response length with offloading is lower than that without offloading at inference time, demonstrating the improved token efficiency of on-device LLMs under our device-cloud collaboration framework.

To further examine whether this early-exit behavior (i.e., short response) generalizes beyond a single model-task setting, we additionally report the response lengths of calling-for-help and independent-reasoning traces for Llama-3.2-3B-Instruct across multiple math reasoning benchmarks. The results are shown in Table 5. As we can see from this table, the calling-for-help traces are consistently shorter than the independent-reasoning traces across all evaluated tasks. Together with Figure 11, which shows the same pattern across both Qwen2.5-1.5B and Llama-3.2-3B on MATH-lighteval, these results indicate that the model learns a consistent routing pattern across different models and tasks: it exits earlier when a problem is beyond its capacity, while completing the full reasoning process independently when it can solve the problem locally.

*Table 5.* Average response length of independent-reasoning and calling-for-help traces for Llama-3.2-3B across different math reasoning benchmarks. Calling-for-help traces are consistently shorter than independent-reasoning traces, suggesting that the early-exit routing pattern generalizes across tasks.

| Type | MATH-lighteval | MATH-500 | AMC23 | MinervaMath | AGI-Eval-Math | Avg. |
|---|---|---|---|---|---|---|
| Independent reasoning | 327 | 385 | 402 | 582 | 453 | 430 |
| Calling for help | 109 | 181 | 195 | 253 | 289 | 205 |

### A.4. Analysis of Call-for-Help Traces

One potential concern is that the on-device model may learn a shortcut policy: instead of attempting to reason about the input query, it may immediately emit the `<unknown>` token based on superficial features of the prompt, such as length or apparent complexity. Such behavior would reduce the learned policy to a shallow difficulty classifier rather than a genuine internal reasoning-based routing mechanism. In this section, we provide additional analysis to rule out this possibility.

We first examine whether the on-device model performs substantive reasoning before invoking cloud assistance. Specifically, we analyze the call-for-help traces produced by Qwen2.5-1.5B on the MATH-lighteval test set. We measure both the number of explicit reasoning steps and the number of generated tokens before the model calls for help. Here, a reasoning step refers to an explicit segment in the model output formatted as `Step 1`, `Step 2`, and so on before the `<unknown>` token. As shown in Table 6, the on-device model generates an average of 4.92 reasoning steps and 185 words before calling for help, with a median of 5 steps and 179 words. Moreover, only 0.37% of call-for-help traces invoke assistance within 30 words, and none do so within 10 words. This indicates that premature surrender is extremely rare. Therefore, the learned policy is unlikely to be merely a shallow classifier that immediately routes difficult-looking queries to the cloud. Instead, the model typically attempts internal reasoning first and calls for help only after progressing through several intermediate steps, supporting the intended design of reasoning-aware internal routing.

*Table 6.* Quantitative analysis of call-for-help traces produced by Qwen2.5-1.5B on the MATH-lighteval test set. The model rarely calls for help immediately; instead, it usually performs several reasoning steps before invoking cloud assistance.

| Metric | Value |
|---|---|
| Total traces | 5000 |
| Call-for-help traces | 1607 |
| Mean reasoning steps before help | 4.92 |
| Median reasoning steps before help | 5 |
| Mean words before help | 185 |
| Median words before help | 179 |
| Call for help within 30 words | 0.37% (6/1607) |
| Call for help within 10 words | 0 |

In addition, the representative examples in Appendix G show that even when the model exits relatively early, it typically performs intermediate reasoning before producing the `<unknown>` token.

### A.5. Learning Dynamics of the Router in Task-Tuning&Router

This section examines the training trajectory of the router module under the two-stage approach baseline, i.e., Task-Tuning&Router. We report the accuracy curve over router-training epochs to illustrate its convergence behavior and confirm that the baseline was fully optimized before comparison. The results in Figure 12 are averaged over two routers trained for Llama-3.2-3B-Instruct and Qwen2.5-1.5B-Instruct models on MATH-lighteval dataset. As we can see in this figure, the router achieves high training accuracy and exhibits clear convergence, confirming that the two-stage baseline was fully optimized. However, the testing accuracy remains lower, indicating poor generalization. Although the router can fit the training distribution well, its predictions on unseen data remain unreliable because surface patterns of the queries do not faithfully reflect the true difficulty of the underlying tasks. As shown in Figure 13, the router's task performance improves as training progresses. Nonetheless, the improvement is limited, further reflecting the inherent difficulty of predicting problem difficulty from surface-level query features alone.

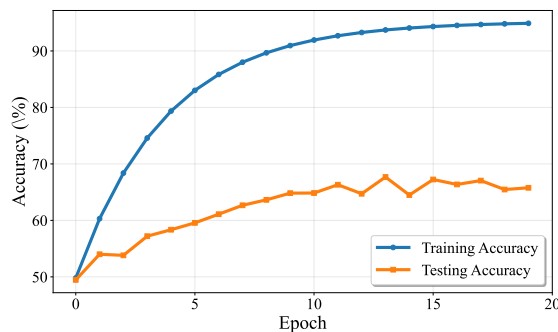
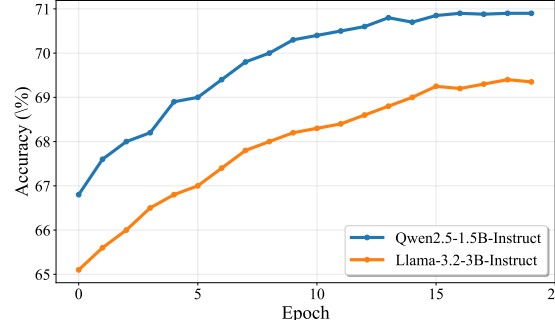

*Figure 12.* Training dynamics of the router in Task-Tuning&Router baseline, averaged over routers for Llama-3.2-3B-Instruct and Qwen2.5-1.5B-Instruct models on MATH-lighteval dataset.

*Figure 13.* Performance of Task-Tuning&Router baseline versus the training epochs of the router under a 30% call-for-help budget with Llama-3.2-3B-Instruct and Qwen2.5-1.5B-Instruct models on MATH-lighteval dataset.

### A.6. More Baselines

To further demonstrate the advantages of our proposed method, we compare it with two recently proposed schemes: AutoMix (Aggarwal et al., 2024) and Router-R1 (Zhang et al., 2026). AutoMix requires the small model to first generate an answer and then self-verify its output before deciding whether to retain the local response or offload the query to the cloud LLM. As a result, AutoMix incurs two local-model passes for each query. Router-R1 is a reasoning-based router originally designed for centralized coordination in multi-agent systems. We adapt Router-R1 to the device-cloud collaboration setting by treating the on-device LLM and the cloud LLM as two agents. We instantiate the router with the same model size as the on-device LLM.

*Table 7.* Testing accuracy (%) of our approach and other two baselines with Qwen2.5-1.5B-Instruct and Llama-3.2-3B-Instruct, tuned on MATH-lighteval. Our method maintain the advantage across both models.

| Model | Method | MATH-lighteval | MATH-500 | AMC23 | MinervaMath | AGI-Eval-Math | Avg. |
|---|---|---|---|---|---|---|---|
| | AutoMix | 59.4 | 57.2 | 42.5 | 23.2 | 54.9 | 47.4 |
| Qwen2.5-1.5B | Router-R1 | 73.6 | 74.1 | 52.5 | 37.8 | 70.4 | 61.9 |
| | **Ours** | **80.4** | **81.6** | **57.5** | **40.8** | **73.4** | **66.7** |
| | AutoMix | 55.3 | 47.9 | 35.0 | 32.7 | 50.1 | 44.2 |
| LLaMA-3.2-3B | Router-R1 | 72.7 | 64.9 | 47.5 | 39.7 | 59.2 | 56.8 |
| | **Ours** | **79.5** | **68.6** | **52.5** | **43.4** | **64.5** | **61.7** |

The results in Table 7 show that our method consistently outperforms both AutoMix and Router-R1 across all evaluated math reasoning benchmarks and both local models. In particular, compared with Router-R1, which relies on an external reasoning-based router, our method achieves higher average accuracy by 4.8% on Qwen2.5-1.5B and 4.9% on LLaMA-3.2-3B. These gains suggest that embedding the routing decision into the model's internal reasoning process provides a more effective mechanism for device-cloud collaboration than using a separately trained external router. Moreover, unlike Router-R1, our scheme does not require deploying an additional reasoning router, thereby eliminating the associated deployment overhead and reducing memory consumption.

### A.7. Switching Cloud LLM at Inference Time

In our main experiments, we use DeepSeek-R1 as the cloud LLM. A natural question is whether the learned routing behavior is overly tied to this particular cloud model. In particular, if the cloud model is replaced by a weaker model at inference time, it is important to understand whether the on-device model still makes appropriate call-for-help decisions.

To evaluate this, we conduct a cloud-model swapping experiment at inference time. Specifically, we train our method using DeepSeek-R1 as the cloud LLM, and then replace it at inference time with a weaker alternative, Qwen3-30B-A3B-Instruct. We use Qwen2.5-1.5B as the on-device model and keep the offloading ratio fixed at 30%. This setting directly tests whether the learned routing policy remains effective when the cloud model used at inference differs from the one used

during training.

*Table 8.* Performance when switching the cloud LLM at inference time. The on-device model is Qwen2.5-1.5B, and the offloading ratio is fixed at 30%. Our method is trained with DeepSeek-R1 as the cloud LLM and evaluated with either DeepSeek-R1 or Qwen3-30B-A3B-Instruct as the inference-time cloud model.

| Setting | Cloud LLM | MATH-lighteval | MATH-500 | AMC23 | MinervaMath | AGI-Eval-Math | Avg. |
|---|---|---|---|---|---|---|---|
| Cloud only | DeepSeek-R1 | 98.4 | 97.3 | 97.5 | 80.9 | 94.7 | 93.8 |
| Cloud only | Qwen3-30B-A3B | 95.6 | 93.8 | 97.5 | 77.6 | 93.3 | 91.6 |
| Ours | DeepSeek-R1 | 80.4 | 81.6 | 57.5 | 40.8 | 73.4 | 66.7 |
| Ours | Qwen3-30B-A3B | 79.5 | 80.4 | 57.5 | 40.1 | 73.2 | 66.1 |

As shown in Table 8, replacing DeepSeek-R1 with Qwen3-30B-A3B-Instruct slightly reduces the cloud-only average performance from 93.8 to 91.6, confirming that Qwen3-30B-A3B-Instruct is a weaker cloud model in this setting. However, the performance of our method decreases only marginally, from 66.7 to 66.1 on average. This suggests that the learned routing behavior is not highly sensitive to the exact identity of the cloud LLM, as long as the replacement cloud model remains substantially stronger than the on-device model.

Overall, the router primarily learns when the on-device model is unlikely to solve a prompt independently and when external assistance is beneficial. When the cloud model is swapped at inference time, the same routing policy can remain appropriate if the new cloud model preserves the same broad capability gap over the on-device model.

## B. Further Justification on Our Unified RL Methodology for Constrained Optimization

This section provides a mathematical justification for the proposed hierarchical reward and prompt filtering mechanisms.

**Hierarchical Reward:** We formulate the training objective as a constrained reward maximization problem (Section 3.1):

$$\max_{\boldsymbol{\theta}} \quad \mathbb{E}_{\boldsymbol{x} \sim \mathcal{D}}[R(\boldsymbol{\theta}, \boldsymbol{x})] := \mathbb{E}_{\boldsymbol{x} \sim \mathcal{D}} \mathbb{E}_{\boldsymbol{y}^{\theta} \sim \pi_{\theta}(\boldsymbol{x})}[r(\boldsymbol{x}, \boldsymbol{y})]$$
$$\text{s.t.} \quad \mathbb{E}[\mathbf{1}\{\boldsymbol{y} \sim (\pi_{\theta}, \pi_c)\}] \leq \rho\, \mathbb{E}[\mathbf{1}\{\boldsymbol{y} \sim \pi_{\theta}\}]. \tag{6}$$

Since each input is either solved locally or offloaded to the cloud, we have

$$\mathbb{E}[\mathbf{1}\{\boldsymbol{y} \sim \pi_{\theta}\}] + \mathbb{E}[\mathbf{1}\{\boldsymbol{y} \sim (\pi_{\theta}, \pi_c)\}] = |\mathcal{D}|. \tag{7}$$

Using this identity, the constraint in Problem (6) can be equivalently rewritten as

$$\mathbb{E}[\mathbf{1}\{\boldsymbol{y} \sim (\pi_{\theta}, \pi_c)\}] \leq \frac{\rho}{1+\rho} |\mathcal{D}|. \tag{8}$$

Applying the Lagrangian relaxation (Schulman et al., 2015; 2017), we convert the constrained problem into an unconstrained one:

$$\max_{\boldsymbol{\theta}} \quad \mathbb{E}_{\boldsymbol{x} \sim \mathcal{D}} \mathbb{E}_{\boldsymbol{y}^{\theta} \sim \pi_{\theta}(\boldsymbol{x})}[r(\boldsymbol{x}, \boldsymbol{y})] - \gamma \left( \mathbb{E}[\mathbf{1}\{\boldsymbol{y} \sim (\pi_{\theta}, \pi_c)\}] - \frac{\rho}{1+\rho}|\mathcal{D}| \right), \tag{9}$$

where $\gamma \geq 0$ is a tunable penalty coefficient.

Dropping the constant term $\gamma \frac{\rho}{1+\rho}|\mathcal{D}|$, which does not affect optimization, the objective simplifies to

$$\max_{\boldsymbol{\theta}} \mathbb{E}_{\boldsymbol{x} \sim \mathcal{D}} \left[ \mathbb{E}_{\boldsymbol{y}^{\theta} \sim \pi_{\theta}(\boldsymbol{x})}[r(\boldsymbol{x}, \boldsymbol{y}) - \gamma\, \mathbf{1}\{\boldsymbol{y} \sim (\pi_{\theta}, \pi_c)\}] \right]. \tag{10}$$

Therefore, the above objective is equivalent to optimizing a modified reward function

$$\tilde{r}(\boldsymbol{x}, \boldsymbol{y}) = \begin{cases} r(\boldsymbol{x}, \boldsymbol{y}) - \gamma, & \text{if } \boldsymbol{y} \sim (\pi_{\theta}, \pi_c), \\ r(\boldsymbol{x}, \boldsymbol{y}), & \text{otherwise.} \end{cases} \tag{11}$$

$$\max_{\boldsymbol{\theta}} \mathbb{E}_{\boldsymbol{x} \sim \mathcal{D}} \left[ \mathbb{E}_{\boldsymbol{y}^{\theta} \sim \pi_{\theta}(\boldsymbol{x})} \left[ \tilde{r}(\boldsymbol{x}, \boldsymbol{y}) \right] \right]. \tag{12}$$

When a rule-based correctness reward is used for $r(\boldsymbol{x}, \boldsymbol{y})$, any correct answer generated via cloud assistance incurs an additional penalty $\gamma$. As a result, the effective reward becomes *hierarchical*, assigning a lower reward to cloud-assisted correct solutions than to correct solutions produced independently by the on-device model.

**Prompt Filtering.** The hierarchical reward implicitly penalizes excessive reliance on the cloud LLM and is sufficient from a static optimization perspective. However, the post-training of LLMs differs slightly from standard constrained optimization and even from conventional reinforcement learning, due to the data generation and policy update mechanism.

LLM post-training typically follows a highly *exploitative procedure*: for each prompt, multiple responses are sampled from the current policy, and only the highest-reward responses are reinforced. As a result, the training data distribution is tightly coupled to the current policy, and future rollouts are strongly influenced by past optimization. This bias toward exploitation is further exacerbated by the extremely large action space of language models, which makes systematic exploration difficult. In contrast, conventional reinforcement learning explicitly balances exploration and exploitation through stochastic policies and state transitions in a Markov decision process. The policy iteratively explores new states while optimizing long-term returns. Such exploration mechanisms are largely absent in standard LLM post-training.

To facilitate analysis, we rewrite Problem 12 as

$$\begin{aligned} \max_{\boldsymbol{\theta}} \quad & \mathbb{E}_{\boldsymbol{x} \sim \mathcal{D}^1} \left[ \mathbb{E}_{\boldsymbol{y}^{\theta} \sim \pi_{\theta}(\boldsymbol{x})} \left[ r(\boldsymbol{x}, \boldsymbol{y}) \right] \right] \\ & + \mathbb{E}_{\boldsymbol{x} \sim \mathcal{D}^2} \left[ \mathbb{E}_{\boldsymbol{y}^{\theta} \sim \pi_{\theta}(\boldsymbol{x})} \left[ r(\boldsymbol{x}, \boldsymbol{y}) - \gamma \, \mathbf{1}\{\boldsymbol{y} \sim (\pi_{\theta}, \pi_c)\} \right] \right], \end{aligned} \tag{13}$$

where $\mathcal{D}^1$ and $\mathcal{D}^2$ denote prompts that are solvable and unsolvable by the on-device LLM $\pi_{\theta}$, respectively. Importantly, this partition is *policy-dependent* rather than static, since the solvability of a prompt depends on the current policy. A more rigorous formulation is therefore

$$\begin{aligned} \max_{\boldsymbol{\theta}} \quad & \mathbb{E}_{\boldsymbol{x} \sim \mathcal{D}^1(\theta)} \left[ \mathbb{E}_{\boldsymbol{y}^{\theta} \sim \pi_{\theta}(\boldsymbol{x})} \left[ r(\boldsymbol{x}, \boldsymbol{y}) \right] \right] \\ & + \mathbb{E}_{\boldsymbol{x} \sim \mathcal{D}^2(\theta)} \left[ \mathbb{E}_{\boldsymbol{y}^{\theta} \sim \pi_{\theta}(\boldsymbol{x})} \left[ r(\boldsymbol{x}, \boldsymbol{y}) - \gamma \, \mathbf{1}\{\boldsymbol{y} \sim (\pi_{\theta}, \pi_c)\} \right] \right], \end{aligned} \tag{14}$$

where $\mathcal{D}^1(\theta) \cup \mathcal{D}^2(\theta) = \mathcal{D}$.

If the sampled prompts are primary from $\mathcal{D}^2(\theta)$ in the beginning of the training, which exceed the on-device model's capability, invoking the cloud model becomes the locally optimal action even in the presence of a Lagarangian regularizer (i.e., hierarchical reward). Due to the exploitative nature of LLM post-training, this behavior reinforces itself and biases the policy toward cloud invocation, causing the induced prompt distribution to collapse. Formally,

$$\Pr_{x \sim \mathcal{D}} \left[ x \in \mathcal{D}^1(\theta) \right] \to 0, \qquad \Pr_{x \sim \mathcal{D}} \left[ x \in \mathcal{D}^2(\theta) \right] \to 1. \tag{15}$$

Consequently, the model will rarely generate independent reasoning trajectories in the following rollouts, even when later exposed to prompts that are, in principle, solvable independently, thereby hindering effective learning of local problem-solving capabilities.

To prevent this collapse, we introduce prompt filtering. Specifically, during each policy update, training prompts are sampled from both $\mathcal{D}^1(\theta)$ and $\mathcal{D}^2(\theta)$. Prompt filtering constructs a controlled training distribution

$$\tilde{d}_{\theta}(x) = \frac{1}{\rho + 1} \, d(x \mid x \in \mathcal{D}^1(\theta)) + \frac{\rho}{\rho + 1} \, d(x \mid x \in \mathcal{D}^2(\theta)), \tag{16}$$

where $d(x)$ denotes the distribution over prompts in $\mathcal{D}$, $\tilde{d}_{\theta}(x)$ represents the distribution over data used for each training iteration, and $\rho$ is determined by the cloud usage budget defined in 6. This guarantees that the training data always contains prompts that admit independent reasoning, while maintaining exposure to prompts that require cloud assistance.

Overall, hierarchical reward and prompt filtering jointly enable stable optimization of local reasoning and cloud cooperation, while the tunable hierarchical reward makes the desired trade-off between local reasoning and cloud assistance attainable.

# C. Comparison of Our Proposed Methodology with GRPO and Dr. GRPO

Our work targets a device-cloud collaboration setting, where the key challenge lies not only in improving on-device reasoning performance but also in jointly optimizing the trade-off between local reasoning and cloud invocation under budget constraints. To tackle this, we introduce hierarchical rewards and propose the GAPG algorithm. In this section, we compare the algorithmic distinctions between GAPG and GRPO and Dr. GRPO and explain why these methods are not well suited for the device-cloud collaboration scenario.

## C.1. Discussion on GRPO

The training objective of GRPO is given by

$$
\begin{aligned}
\max_{\boldsymbol{\theta}} \; & \mathbb{E}_{\boldsymbol{x}\sim\mathcal{D},\{\boldsymbol{y}_i\}_{i=1}^{G}\sim\pi_{\theta_{\text{old}}}(\boldsymbol{x})} \frac{1}{G}\sum_{i=1}^{G}\frac{1}{|\boldsymbol{y}_i|}\sum_{t=1}^{|\boldsymbol{y}_i|}\left\{\min\left[\frac{\pi_\theta\left(y_{i,t}\mid\boldsymbol{x},\boldsymbol{y}_{i,<t}\right)}{\pi_{\theta_{\text{old}}}\left(y_{i,t}\mid\boldsymbol{x},\boldsymbol{y}_{i,<t}\right)}A_{i,t},\right.\right. \\
& \left.\left.\text{clip}\left(\frac{\pi_\theta\left(y_{i,t}\mid\boldsymbol{x},\boldsymbol{y}_{i,<t}\right)}{\pi_{\theta_{\text{old}}}\left(y_{i,t}\mid\boldsymbol{x},\boldsymbol{y}_{i,<t}\right)},1-\varepsilon,1+\varepsilon\right)A_{i,t}\right]-\beta\mathbb{D}_{\text{KL}}\left[\pi_\theta||\pi_{\text{ref}}\right]\right\}, \\
& \text{where } A_{i,t}=(r_i-\text{mean}(\{r_i\}_{i=1}^{G}))/\text{std}(\{r_i\}_{i=1}^{G}).
\end{aligned}
\tag{17}
$$

In particular, $\pi_{\theta_{\text{old}}}$ denotes the stale policy used for sampling, and $\pi_{\text{ref}}$ is the reference model, typically set to the initial policy to penalize deviations from the starting point. The hyperparameters $\epsilon$ and $\beta$ control the clipping range and the strength of KL regularization, respectively. Note that the trajectory-level normalized relative advantage defined in (17) is assigned to each token, i.e., $A_{i,t}=A_i$.

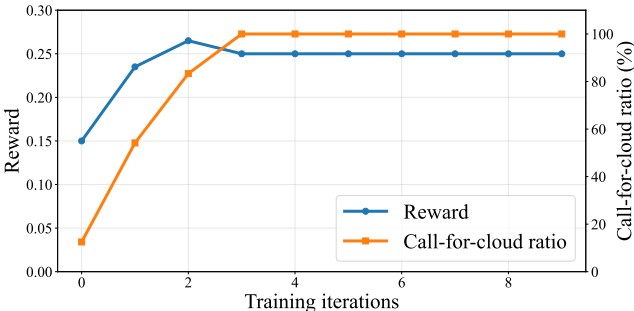

*Figure 14.* Rewards and call-for-cloud ratios over training iterations. The reward converges to the coordination reward, while the on-device LLM collapses to a degenerate policy that always invokes the cloud model (100% call-for-cloud ratio).

The misalignment between normalized advantages and true rewards makes GRPO not a good fit for our setup. Specifically, the normalized advantage used in GRPO can distort the actual value of responses, which is problematic in our setting with hierarchical rewards. For instance, consider two groups of responses to the same prompt: Group A with rewards $[\alpha_a,0,0,0,0,0,0,0]$ and Group B with rewards $[\alpha_c,0,0,0,0,0,0,0]$, where $\alpha_a$ and $\alpha_c$ ($\alpha_a>\alpha_c$) correspond to producing a correct answer independently and invoking the cloud model for assistance, respectively. After normalization, the advantages of the highest-reward responses in both groups are the same, despite the clear superiority of the response with reward $\alpha_a$. This causes the model to treat both responses as equally valuable, failing to recognize the greater merit of the correct answer through independent local reasoning.

For illustration, we apply GRPO to optimize the on-device LLM $\pi_\theta$ under the proposed collaboration-aware hierarchical rewards (Section 3.2). Specifically, we take Qwen2.5-3B-Instruct as the on-device model and DeepSeek-R1 as the cloud model. As the evaluation benchmark, we adopt the Countdown task (Pan et al., 2025), a mathematical puzzle in which players must combine a given set of numbers using the four basic arithmetic operations $(+,-,\times,\div)$ to reach a specific target number. For instance, given the numbers 75, 6, 2, and 3 with a target of 152, one valid solution is: $(75\div3)\times6+2=152$. This task provides a wide range of problems with varying levels of difficulty, making it particularly suitable for the on-device and cloud LLMs collaboration settings. The reward parameters in Section 3.2 are set as $\alpha_a=1$ and $\alpha_c=0.25$. Figure 14 illustrates the evolution of the training reward and the call-for-cloud ratio with respect to training iterations. As shown in Figure 14, the model trained with GRPO converges to a low-quality policy that frequently calls for assistance from

the cloud LLM. This occurs because the call-for-help pattern is easier to learn than independent reasoning. Moreover, these two behaviors are in a competitive relationship and are mutually exclusive. When they are treated equally in the advantage computation, the model tends to collapse to a degenerate policy that relies almost exclusively on cloud assistance.

### C.2. Discussion on Dr. GRPO

Liu et al. (2025) proposed Dr. GRPO where the objective is given by

$$
\max_{\boldsymbol{\theta}} \ \mathbb{E}_{\boldsymbol{x} \sim \mathcal{D}} \frac{1}{G} \sum_{i=1}^{G} \sum_{t=1}^{|\boldsymbol{y}_i|} \left\{ \min \left[ \frac{\pi_\theta \left( y_{i,t} \mid \boldsymbol{x}, \boldsymbol{y}_{i,<t} \right)}{\pi_{\theta_{\text{old}}} \left( y_{i,t} \mid \boldsymbol{x}, \boldsymbol{y}_{i,<t} \right)} A_{i,t}, \text{clip} \left( \frac{\pi_\theta \left( y_{i,t} \mid \boldsymbol{x}, \boldsymbol{y}_{i,<t} \right)}{\pi_{\theta_{\text{old}}} \left( y_{i,t} \mid \boldsymbol{x}, \boldsymbol{y}_{i,<t} \right)}, 1 - \varepsilon, 1 + \varepsilon \right) A_{i,t} \right] \right\}, \quad (18)
$$
$$
\text{where } A_{i,t} = (r_i - \text{mean}(\{r_i\}_{i=1}^{G})),
$$

where the notations are the same as those of GRPO.

Although Dr. GRPO does not suffer from the misalignment issue observed in GRPO, it is still not well suited for our device-cloud collaboration setting. First, Dr. GRPO maximizes token-level rewards. To illustrate, we omit the clipping operation, i.e., assuming on-policy updates, and rewrite its objective in (18) as $\mathbb{E}_{\boldsymbol{x} \sim \mathcal{D}} \left[ \frac{1}{G} \sum_{i=1}^{G} \sum_{t=1}^{|\boldsymbol{y}_i|} A_{i,t} \right]$. This formulation maximize the sum of individual token rewards across the entire trajectory, which is problematic for device-cloud collaboration. For example, a long response that merely asks for cloud assistance could accumulate a total reward equal to or even higher than a concise, correct on-device answer. This occurs whenever the length of the longer response compensates for its lower per-token reward, thereby misleading the on-device model into favoring low-effort offloading over successful independent reasoning. Second, Dr. GRPO does not account for the constraint on invoking the cloud LLM. Since the assistance-invoking pattern is comparatively easy to learn, the model tends to overuse it, which leads to premature convergence to an undesirable policy. More comparison between Dr. GRPO and our methodology are provided in Section 4 (i.e., Collaborative-Aware Tuning baseline).

## D. Prompt and Hierarchical Reward Details

**Prompt template.** We design two prompt templates that explicitly instruct the model to answer only when confident, as shown in Table 9. Template II provides stricter guidance than Template I and can be applied when the model fails to recognize its knowledge limitations under Template I.

*Table 9.* Prompt templates for training the on-device LLM in the collaborative device–cloud framework. The placeholder *question* and *number* will be replaced with the actual question and an appropriate number during training.

| |
|---|
| **Template I.** *System prompt:* You are a helpful assistant. You first think about the reasoning process in your mind and then provide the user with the answer. Show all your reasoning in `<think> </think>` tags. And return the final answer in `<answer> </answer>` tags at the end. If you did not find a solution after a thorough reasoning process, you can ask for external assistance at the end, for example, `<unknown>` I need external assistance `</unknown>`. User: *question*. Let me solve this step by step. |
| **Template II.** *System prompt:* You are given a math problem. Solve it step by step. Organize your thoughts using this format: Step 1: ..., Step 2: ..., Step 3: ..., and so on. Put your final answer within `\boxed{}`. If you cannot solve the problem after *number* reasoning steps, stop reasoning and return: `<unknown>` I need external assistance `</unknown>`. User: *question*. Let's think step by step. |

In our experiments, we use Template I for the Countdown task and Template II for the MATH-lighteval task. The tunable hyperparameter *number* is set to 6.

**Hierarchical Rewards.** All possible rewards discussed in Section 3.2 are summarized in the following equation, which integrates two components of our collaboration-aware rewards, *accuracy* and *coordination* rewards, to jointly encourage

independent problem-solving and judicious cloud offloading:

$$r(\boldsymbol{x}, \boldsymbol{y}) = \begin{cases} \alpha_a, & \text{if } \boldsymbol{y} = \boldsymbol{y}^\theta \text{ contains the correct answer} \\ \alpha_c, & \text{if } \boldsymbol{y} = [\boldsymbol{y}^\theta, \boldsymbol{y}^c] \text{ contains the correct answer} \\ 0, & \text{otherwise.} \end{cases} \tag{19}$$

## E. Details on Datasets and Hyperparameters

**Details on Datasets.** We fine-tune the on-device LLM on two datasets, Countdown (Pan et al., 2025) and MATH-lighteval (Hendrycks et al., 2021), respectively. The Countdown task is an arithmetic puzzle where the model must combine a given set of numbers using basic arithmetic operations $(+, -, \times, \div)$ to reach a specified target number. We randomly select 6,000 problems from the Countdown dataset, using 5,000 for training and 1,000 for testing. The MATH-lighteval dataset comprises 12,500 problems drawn from mathematics competitions, covering topics such as algebra, geometry, counting and probability, number theory, and precalculus. We use 7,500 samples for training and 5,000 for testing. Additionally, we evaluate the models fine-tuned on the MATH-lighteval dataset against four widely used mathematical benchmarks: MATH-500 (Hendrycks et al., 2021), AMC23 (Lewkowycz et al., 2022), AGI-Eval-Math (Zhong et al., 2024), and MinervaMath (Li et al., 2024a).

**Details on Hyperparameters.** Unless stated otherwise, the hyperparameters used for the Countdown task under the Qwen2.5-3B-Instruct and MATH-lighteval task under Llama-3.2-1B-Instruct, Qwen2.5-1.5B-Instruct, and Llama-3.2-3B-Instruct are as follows.

*Table 10.* The hyperparameters for the Countdown task under the Qwen2.5-3B-Instruct model and the MATH-lighteval task under Llama-3.2-1B-Instruct, Qwen2.5-1.5B-Instruct, and Llama-3.2-3B-Instruct models.

| Hyperparameter | Countdown & Qwen | Math-lighteval & LLaMA-3.2-1B/Qwen2.5-1.5B/LLaMA-3.2-3B |
|---|---|---|
| Batch size $|\mathcal{D}_b|$ | 32 | 128 |
| Group size $G$ | 8 | 8 |
| Max prompt length | 256 | 1024 |
| Max response length | 720 | 1024 |
| Learning rate $\eta$ | 5e-6 | 2e-6 |
| Total training steps $S$ | 800 | 800 |
| Sampling temperature for training | 1.0 | 1.0 |
| Sampling temperature for evaluation | 0 | 0 |
| Accuracy reward | 1 | 1 |
| Coordination reward | 0.25 | 0.2/0.3/0.6 |
| Cloud-device usage ratio $\rho$ | 3/7 | 3/7 |

## F. Proof of Proposition 3.1

### F.1. Part I: Unbiasedness

Recall the estimator

$$\widehat{\nabla_\theta R}(\boldsymbol{\theta}, \boldsymbol{x}) = \frac{G}{G-1} \frac{1}{G} \sum_{i=1}^{G} \left[ \nabla_\theta \log \pi_\theta(\boldsymbol{y}_i^\theta \mid \boldsymbol{x}) \right] (r_i - \bar{r}), \quad \bar{r} = \frac{1}{G} \sum_{j=1}^{G} r_j.$$

Because each $\boldsymbol{y}_i^\theta \sim \pi_\theta(\boldsymbol{x})$ and $\int \pi_\theta(\boldsymbol{y} \mid \boldsymbol{x})\,d\boldsymbol{y} = 1$, the following identity holds for every $i$:

$$
\begin{aligned}
\mathbb{E}_{\boldsymbol{y}_i^\theta \sim \pi_\theta(\boldsymbol{x})}\!\left[\nabla_\theta \log \pi_\theta(\boldsymbol{y}_i^\theta \mid \boldsymbol{x})\right] &= \int \pi_\theta(\boldsymbol{y} \mid \boldsymbol{x})\nabla_\theta \log \pi_\theta(\boldsymbol{y} \mid \boldsymbol{x})\,d\boldsymbol{y} \\
&= \int \nabla_\theta \pi_\theta(\boldsymbol{y} \mid \boldsymbol{x})\,d\boldsymbol{y} \\
&= \nabla_\theta \int \pi_\theta(\boldsymbol{y} \mid \boldsymbol{x})\,d\boldsymbol{y} \\
&= 0.
\end{aligned}
\tag{20}
$$

Following the log-likelihood trick, we have

$$
\begin{aligned}
\mathbb{E}_{\boldsymbol{y}_i^\theta \sim \pi_\theta(\boldsymbol{x})}\!\left[\nabla_\theta \log \pi_\theta(\boldsymbol{y}_i^\theta \mid \boldsymbol{x})\,r(\boldsymbol{x}, \boldsymbol{y}_i)\right] &= \int \pi_\theta(\boldsymbol{y}_i^\theta \mid \boldsymbol{x})\nabla_\theta \log \pi_\theta(\boldsymbol{y}^\theta \mid \boldsymbol{x})\,r(\boldsymbol{x}, \boldsymbol{y})\,d\boldsymbol{y} \\
&= \int \nabla_\theta \pi_\theta(\boldsymbol{y}^\theta \mid \boldsymbol{x})\,r(\boldsymbol{x}, \boldsymbol{y})\,d\boldsymbol{y} \\
&= \nabla_\theta \int \pi_\theta(\boldsymbol{y}^\theta \mid \boldsymbol{x})\,r(\boldsymbol{x}, \boldsymbol{y})\,d\boldsymbol{y} \\
&= \nabla_\theta \mathbb{E}_{\boldsymbol{y}^\theta \sim \pi_\theta(\boldsymbol{x})}\!\left[r(\boldsymbol{x}, \boldsymbol{y})\right] \\
&= \nabla_\theta R(\boldsymbol{\theta}, \boldsymbol{x}).
\end{aligned}
\tag{21}
$$

Using linearity of expectation, we have

$$
\mathbb{E}\!\left[\widehat{\nabla_\theta R}(\boldsymbol{\theta}, \boldsymbol{x})\right] = \frac{G}{G-1} \frac{1}{G} \sum_{i=1}^{G} \mathbb{E}\!\left[\nabla_\theta \log \pi_\theta(\boldsymbol{y}_i^\theta \mid \boldsymbol{x})\left(r_i - \bar{r}\right)\right].
$$

Because the $G$ terms are identically distributed, replace the sum by a single expectation:

$$
\begin{aligned}
\mathbb{E}\!\left[\widehat{\nabla_\theta R}(\boldsymbol{\theta}, \boldsymbol{x})\right] &= \frac{G}{G-1} \mathbb{E}\!\left[\nabla_\theta \log \pi_\theta(\boldsymbol{y}_1^\theta \mid \boldsymbol{x})\left(r_1 - \bar{r}\right)\right] \\
&= \frac{G}{G-1} \mathbb{E}\!\left[\nabla_\theta \log \pi_\theta(\boldsymbol{y}_1^\theta \mid \boldsymbol{x})\left((1 - \tfrac{1}{G})r_1 - \tfrac{1}{G}\sum_{j \neq 1} r_j\right)\right] \\
&= \mathbb{E}\!\left[\nabla_\theta \log \pi_\theta(\boldsymbol{y}_1^\theta \mid \boldsymbol{x})r_1\right] - \frac{1}{G-1} \sum_{j \neq 1} \mathbb{E}\!\left[\nabla_\theta \log \pi_\theta(\boldsymbol{y}_1^\theta \mid \boldsymbol{x})r_j\right].
\end{aligned}
\tag{22}
$$

For the term with $r_1$, following (21), we have

$$
\mathbb{E}\!\left[\nabla_\theta \log \pi_\theta(\boldsymbol{y}_1^\theta \mid \boldsymbol{x})\,r_1\right] = \nabla_\theta R(\boldsymbol{\theta}, \boldsymbol{x}).
\tag{23}
$$

For the term with $r_j, j \neq 1$, the random variables $\nabla_\theta \log \pi_\theta(\boldsymbol{y}_1^\theta \mid \boldsymbol{x})$ and $r_j$ are independent, giving

$$
\begin{aligned}
\mathbb{E}\!\left[\nabla_\theta \log \pi_\theta(\boldsymbol{y}_1^\theta \mid \boldsymbol{x})\,r_j\right] &= \mathbb{E}[r_j] \cdot \mathbb{E}\!\left[\nabla_\theta \log \pi_\theta(\boldsymbol{y}_1^\theta \mid \boldsymbol{x})\right] \\
&= 0,
\end{aligned}
\tag{24}
$$

where the second inequality comes from (20). Plugging (23) and (24) into (22) gives rise to

$$
\mathbb{E}\!\left[\widehat{\nabla_\theta R}(\boldsymbol{\theta}, \boldsymbol{x})\right] = \nabla_\theta R(\boldsymbol{\theta}, \boldsymbol{x}).
$$

This completes the proof of Part I.

## F.2. Part II: Variance Property

The proof of the Part II relies on the following lemma and its proof is delayed to Appendix F.3

**Lemma F.1.** Let $s_i = \nabla_\theta \log \pi_\theta(y_i^\theta \mid x)$. Then the covariance matrix of the group-level policy gradient estimator $\widehat{\nabla_\theta R}(\theta, x)$ defined in (2) satisfies

$$\text{Cov}\left(\widehat{\nabla_\theta R}(\theta, x)\right) = \frac{1}{G} A + \frac{\sigma_r^2}{G(G-1)} S - \frac{G-2}{G(G-1)} mm^\top, \tag{25}$$

where $A = \mathbb{E}\left[s_1 s_1^\top (r_1 - \mu)^2\right]$, $S = \mathbb{E}[s_1 s_1^\top]$, $m = \nabla_\theta R(\theta, x)$, $\mu = \mathbb{E}[r_1]$, and $\sigma_r^2 = \text{Var}(r_1)$.

With Lemma F.1, $G \geq 2$, and the fact that $A \succeq 0$ and $S \succeq 0$, we have

$$\text{Cov}\left(\widehat{\nabla_\theta R}(\theta, x)\right) \preceq \frac{1}{G} A + \frac{\sigma_r^2}{G(G-1)} S \sim \mathcal{O}(\frac{1}{G}). \tag{26}$$

The results demonstrate that the covariance of the proposed estimator scales inversely with the group size $G$, specifically decaying at a rate of $\mathcal{O}(1/G)$.

In addition, since $A \succeq 0$ and $S \succeq 0$, we have $\text{tr}(A) \geq 0$ and $\text{tr}(S) \geq 0$.

$$\mathbb{E}\left[\left\|\widehat{\nabla_\theta R}(\theta, x) - \nabla_\theta R(\theta, x)\right\|_2^2\right] = \text{tr}\left(\text{Cov}(\widehat{\nabla_\theta R}(\theta, x))\right).$$

Therefore, we have

$$\text{tr}\left(\text{Cov}\left(\widehat{\nabla_\theta R}(\theta, x)\right)\right) \leq \frac{1}{G}\text{tr}(A) + \frac{\sigma_r^2}{G(G-1)}\text{tr}(S) \sim \mathcal{O}\left(\frac{1}{G}\right). \tag{27}$$

This completes the proof of Part II.

## F.3. Proof of Lemma F.1

For the gradient estimator (2), we have

$$\widehat{\nabla_\theta R}(\theta, x) = \frac{1}{G}\sum_{i=1}^{G} s_i \left(\frac{G}{G-1}r_i - \frac{1}{G-1}\sum_{j=1}^{G} r_j\right) = \frac{1}{G}\sum_{i=1}^{G} s_i (r_i - b_i), \tag{28}$$

where $b_i = \frac{1}{G-1}\sum_{j\neq i}^{G} r_j$. Define $Z_i = s_i(r_i - b_i)$, the gradient estimator can be expressed as $\widehat{\nabla_\theta R} = \frac{1}{G}\sum_{i=1}^{G} Z_i$. Hence, we have

$$\text{Cov}\left(\widehat{\nabla_\theta R}(\theta, x)\right) = \frac{1}{G^2}\sum_{i=1}^{G} \text{Cov}(Z_i) + \frac{1}{G^2}\sum_{i\neq j} \text{Cov}(Z_i, Z_j). \tag{29}$$

**1). Characterizing** $\text{Cov}(Z_1)$.

Since $b_1$ depends only on $\{r_j : j \neq 1\}$, it is independent of $(s_1, r_1)$ by independent and identically distributed (i.i.d.) sampling. Also $\mathbb{E}[b_1] = \mu$ and $\text{Var}(b_1) = \sigma_r^2/(G-1)$. Hence, we have

$$\mathbb{E}\left[Z_1 Z_1^\top\right] = \mathbb{E}\left[s_1 s_1^\top ((r_1 - \mu) - (b_1 - \mu))^2\right]$$
$$= \mathbb{E}\left[s_1 s_1^\top (r_1 - b_1)^2\right] \tag{30}$$
$$= \mathbb{E}\left[s_1 s_1^\top (r_1 - \mu)^2\right] + \mathbb{E}\left[s_1 s_1^\top\right] \mathbb{E}\left[(b_1 - \mu)^2\right] - 2\mathbb{E}\left[s_1 s_1^\top (r_1 - \mu)\right] \mathbb{E}[b_1 - \mu].$$

The last term is zero because $\mathbb{E}[b_1 - \mu] = 0$. Moreover, $\mathbb{E}\left[(b_1 - \mu)^2\right] = \frac{\sigma_r^2}{G-1}$. Therefore,

$$\mathbb{E}\left[Z_1 Z_1^\top\right] = A + \frac{\sigma_r^2}{G-1} S.$$

Based on the unbiasedness of the estimator, i.e., $\mathbb{E}[\boldsymbol{Z}_1] = \boldsymbol{m}$, and $\mathrm{Cov}(\boldsymbol{Z}_1) = \mathbb{E}[\boldsymbol{Z}_1 \boldsymbol{Z}_1^\top] - \mathbb{E}[\boldsymbol{Z}_1]\mathbb{E}[\boldsymbol{Z}_1]^\top$, we further have

$$\mathrm{Cov}(\boldsymbol{Z}_1) = \boldsymbol{A} + \frac{\sigma_r^2}{G-1}\boldsymbol{S} - \boldsymbol{m}\boldsymbol{m}^\top. \tag{31}$$

**2) Characterizing** $\mathrm{Cov}(\boldsymbol{Z}_1, \boldsymbol{Z}_2)$

Let $R = \sum_{k \neq 1,2} r_k$ be the sum of the remaining $G - 2$ rewards. Then the baselines are $b_1 = \frac{r_2 + R}{G-1}$ and $b_2 = \frac{r_1 + R}{G-1}$. We expand the product $\mathbb{E}[\boldsymbol{Z}_1 \boldsymbol{Z}_2^\top]$ as follows:

$$\mathbb{E}[\boldsymbol{Z}_1 \boldsymbol{Z}_2^\top] = \mathbb{E}\left[ \boldsymbol{s}_1 \left( r_1 - \frac{r_2 + R}{G-1} \right) \left( r_2 - \frac{r_1 + R}{G-1} \right) \boldsymbol{s}_2^\top \right].$$

Expanding the inner product yields:

$$\begin{aligned}
\mathbb{E}[\boldsymbol{Z}_1 \boldsymbol{Z}_2^\top] &= \mathbb{E}\left[ \boldsymbol{s}_1 \boldsymbol{s}_2^\top \left( r_1 r_2 - \frac{r_1(r_1 + R)}{G-1} - \frac{r_2(r_2 + R)}{G-1} + \frac{(r_1 + R)(r_2 + R)}{(G-1)^2} \right) \right] \\
&= \mathbb{E}[\boldsymbol{s}_1 r_1 \boldsymbol{s}_2^\top r_2] - \frac{\mathbb{E}[\boldsymbol{s}_1 r_1^2 \boldsymbol{s}_2^\top] + \mathbb{E}[\boldsymbol{s}_1 r_1 R \boldsymbol{s}_2^\top]}{G-1} + \frac{\mathbb{E}[\boldsymbol{s}_1 \boldsymbol{s}_2^\top (r_1 r_2 + r_1 R + r_2 R + R^2)]}{(G-1)^2}.
\end{aligned} \tag{32}$$

As $(\boldsymbol{s}_1, r_1)$, $(\boldsymbol{s}_2, r_2)$, and $R$ are mutually independent. Since $\mathbb{E}[\boldsymbol{s}_i] = \boldsymbol{0}$, any term containing $\boldsymbol{s}_1$ or $\boldsymbol{s}_2$ without its corresponding reward $r_1$ or $r_2$ vanishes (e.g., $\mathbb{E}[\boldsymbol{s}_1 r_1^2 \boldsymbol{s}_2^\top] = \mathbb{E}[\boldsymbol{s}_1 r_1^2]\mathbb{E}[\boldsymbol{s}_2^\top] = \boldsymbol{0}$). The only surviving terms are those where $\boldsymbol{s}_1$ is paired with $r_1$ and $\boldsymbol{s}_2$ is paired with $r_2$:

$$\mathbb{E}[\boldsymbol{Z}_1 \boldsymbol{Z}_2^\top] = \mathbb{E}[\boldsymbol{s}_1 r_1]\mathbb{E}[\boldsymbol{s}_2 r_2]^\top + \frac{\mathbb{E}[\boldsymbol{s}_1 r_1]\mathbb{E}[\boldsymbol{s}_2 r_2]^\top}{(G-1)^2}. \tag{33}$$

Note that $\mathbb{E}[\boldsymbol{s}_i r_i] = \mathbb{E}[\boldsymbol{s}_i(r_i - b_i)] = \mathbb{E}[\boldsymbol{Z}_i] = \boldsymbol{m}$ and $\mathrm{Cov}(\boldsymbol{Z}_1, \boldsymbol{Z}_2) = \mathbb{E}[\boldsymbol{Z}_1 \boldsymbol{Z}_2^\top] - \mathbb{E}[\boldsymbol{Z}_1]\mathbb{E}[\boldsymbol{Z}_2]^\top$, we obtain

$$\mathrm{Cov}(\boldsymbol{Z}_1, \boldsymbol{Z}_2) = \left( 1 + \frac{1}{(G-1)^2} \right) \boldsymbol{m}\boldsymbol{m}^\top - \boldsymbol{m}\boldsymbol{m}^\top = \frac{1}{(G-1)^2}\boldsymbol{m}\boldsymbol{m}^\top. \tag{34}$$

**3). Plugging** (31) **and** (34) **into** (29)

By symmetry, $\mathrm{Cov}(\boldsymbol{Z}_i) = \mathrm{Cov}(\boldsymbol{Z}_1)$ for all $i$ and $\mathrm{Cov}(\boldsymbol{Z}_i, \boldsymbol{Z}_j) = \mathrm{Cov}(\boldsymbol{Z}_1, \boldsymbol{Z}_2)$ for all $i \neq j$. (29) thus can be rewritten as

$$\mathrm{Cov}\left( \widehat{\nabla_\theta R}(\boldsymbol{\theta}, \boldsymbol{x}) \right) = \frac{1}{G}\mathrm{Cov}(\boldsymbol{Z}_1) + \frac{G-1}{G}\mathrm{Cov}(\boldsymbol{Z}_1, \boldsymbol{Z}_2). \tag{35}$$

Substituting (31) and (34) into (35) gives rise to

$$\mathrm{Cov}\left( \widehat{\nabla_\theta R}(\boldsymbol{\theta}, \boldsymbol{x}) \right) = \frac{1}{G}\left( \boldsymbol{A} + \frac{\sigma_r^2}{G-1}\boldsymbol{S} - \boldsymbol{m}\boldsymbol{m}^\top \right) + \frac{G-1}{G} \cdot \frac{1}{(G-1)^2}\boldsymbol{m}\boldsymbol{m}^\top. \tag{36}$$

Combining the coefficients of $\boldsymbol{m}\boldsymbol{m}^\top$ yields (25).

## G. Case Study I: Several Representative Call-for-help Patterns

This section presents a case study demonstrating that the on-device LLM does not always complete a full reasoning trace before routing. Instead, the model sometimes halts early when it detects low confidence and calls for help. This study is based on MATH-ligheval task with the Qwen2.5-1.5B-Instruct model.

## Two representative call-for-help patterns

### A short reasoning before calling for help (Early exit)

**Question:** For what real values of $x$ is $-4 < x^4 + 4x^2 < 21$ satisfied? Express your answer in interval notation.
**Response:** **Step 1:** We need to solve the inequality

$$-4 < x^4 + 4x^2 < 21.$$

**Step 2:** Let's start by solving the left part of the inequality,

$$-4 < x^4 + 4x^2.$$

**Step 3:** Let's solve the right part of the inequality,

$$x^4 + 4x^2 < 21.$$

**I need external assistance.**

**Question:** The rightmost non-zero digit in $(1001001)(1010101) + (989899)(1001001) - (1001)(989899) - (1010101)(1001)$ is $a$, and it is followed by $b$ zeroes. Find the ordered pair $(a, b)$.
**Response:** **Step 1:** Let's simplify the given expression step by step.
First, let's simplify each part of the expression separately.
1. Simplify $(1001001)(1010101) + (989899)(1001001)$:

$$\text{Notice that } 1001001 = 1001 \times 1001, \quad \text{and } 1010101 = 1010101.$$

2. Simplify $(1001)(989899) + (1010101)(1001)$:

$$\text{Notice that } 989899 = 1001 \times 9899.$$

**I need external assistance.**

### Long reasoning before calling for help

**Question:** Two positive numbers $p$, $q$ satisfy

$$p + q = pq, \qquad p - q = 7.$$

Compute

$$\frac{1}{\frac{1}{p^2} + \frac{1}{q^2}}$$

in the form $\frac{a+b\sqrt{c}}{d}$, where $a$, $b$ do not both share a factor with $d$ and $c$ is squarefree. Find $a + b + c + d$.
**Response:** **Step 1.** Let's denote the two numbers as $p$ and $q$. We are given two conditions:

$$p + q = pq$$

$$p - q = 7$$

**Step 2.** From the first condition, we can rearrange to get:

$$pq - p - q = 0$$

Adding 1 to both sides:

$$pq - p - q + 1 = 1$$

This can be factored as:

$$(p - 1)(q - 1) = 1$$

**Step 3.** Now we have the system:

$$p + q = pq, \qquad p - q = 7$$

From the second equation, solve for $p$:

$$p = q + 7$$

Substituting into $p + q = pq$,

$$(q + 7) + q = (q + 7)q$$

Simplifying:

$$2q + 7 = q^2 + 7q$$

Rearranging:

$$q^2 + 5q - 7 = 0$$

Using the quadratic formula,

$$q = \frac{-5 \pm \sqrt{25 + 28}}{2}$$

$$q = \frac{-5 \pm \sqrt{53}}{2}$$

Since $q > 0$,

$$q = \frac{-5 + \sqrt{53}}{2}$$

Substitute into $p = q + 7$:

$$p = \frac{-5 + \sqrt{53}}{2} + 7 = \frac{-5 + \sqrt{53} + 14}{2} = \frac{9 + \sqrt{53}}{2}$$

**Step 4.** Now compute:

$$\frac{1}{\frac{1}{p^2} + \frac{1}{q^2}} = \frac{p^2 q^2}{p^2 + q^2}$$

We know $p + q = pq$, so:

$$p^2 + q^2 = (p + q)^2 - 2pq$$

From $p + q = pq$,

$$p^2 + q^2 = p^2 + q^2$$

Now compute:

$$p^2 q^2 = (pq)^2 = p^2 q^2$$

$$p^2 + q^2 = (p + q)^2 - 2pq = p^2 + q^2$$

Thus:

$$\frac{1}{\frac{1}{p^2} + \frac{1}{q^2}} = \frac{p^2 q^2}{p^2 + q^2} = \frac{p^2 q^2}{p^2 + q^2} = 1$$

**I need external assistance.**

---

**Takeaway:** The above examples show that the on-device LLM does not always engage in a long reasoning chain before routing. Instead, early-exit behavior naturally emerges under our methodology.

## H. Case Study II: Comparison of the Responses with Our Algorithm and Baselines

We present representative responses generated by the on-device LLM fine-tuned under different strategies. For illustration, we focus on the Countdown task with the Qwen2.5-3B-Instruct model. Since the three baselines, Task-Tuning Only, Task-Tuning&Naive Offloading, and Task-Tuning&Router, use the same on-device LLM, we report on-device results only under Task-Tuning Only.

---

### Case Study for Countdown & Qwen2.5-3B-Instruct: I

**User:** Using the numbers $[74, 78, 36, 7]$, create an equation that equals 33. You can use basic arithmetic operations $(+, -, *, /)$ and each number can only be used once. Let's think step by step.

#### Cloud LLM

**System prompt:** None
**Response:** I need to use the numbers 74, 78, 36, and 7 to make 33. I can use addition, subtraction, multiplication, and division, and I have to use each number exactly once.
First, I should think about what operations might get me close to 33. Since 33 is a relatively small number compared to 74 and 78, I probably need to subtract or divide the larger numbers to reduce them.
Maybe I can subtract two large numbers to get a small number. For example, $78 - 74 = 4$, which is too small, and I have 36 and 7 left. I need to incorporate all numbers, so that might not work directly.
I could multiply or divide to get closer. 36 is already close to 33, so if I can adjust it with the other numbers, that might work. For instance, $36 - 3$ is 33, but I need to get 3 from somewhere.
I have 7 and the other numbers. $7 - 4 = 3$, but I need to get 4 from somewhere. If I use 78 and 74, $78 - 74 = 4$, but then I have used 78 and 74 to get 4, and I have 36 and 7 left. If I do $7 - 4 = 3$, but I need to use 36. So perhaps $36 - (7 - (78 - 74))$, but that would be $36 - (7 - 4) = 36 - 3 = 33$. But I have used all numbers: 74, 78, 7, 36. Let me write that as an equation.

$$36 - (7 - (78 - 74)) = 36 - (7 - 4) = 36 - 3 = 33. \quad \text{Perfect!}$$

But I need to make sure that the operations are clear and that I use each number once. In this case, I have used 78, 74, 7, and 36. The expression is $36 - (7 - (78 - 74))$.
I should write it properly. Since subtraction is not associative, I need to use parentheses to ensure the order of operations.
So, the equation is:

$$36 - (7 - (78 - 74)) = 33$$

I can simplify the inner part: $78 - 74 = 4$, then $7 - 4 = 3$, then $36 - 3 = 33$. Yes.
I think this works. But let me see if there are other ways, but this seems straightforward.
The problem says "create an equation", so I should present it as an equation.

So, using the numbers, I have:

$$36 - (7 - (78 - 74)) = 33$$

But I used the numbers in the order: 36, 7, 78, 74. And I used each once.
I think this is correct.
**Verdict:** Correct Answer

## On-device LLM under Task-Tuning Only

**System prompt:** You are a helpful assistant. You first think about the reasoning process in your mind and then provide the user with the answer. Show all your reasoning in `<think> </think>` tags. And return the final answer in `<answer> </answer>` tags at the end.

**Response:** `<think>` First, we notice that we have four numbers $[74, 78, 36, 7]$. We need to create an expression using these numbers exactly once that results in 33. Starting with multiplication $74 * 36 = 2664$, which is too large. The quotient when 74 is divided by 36 is approximately 2.0556. Multiplying 2.0556 by 78 gives us approximately 159.9968, still far from 33. Now, let's try using 78 and 36 together. Dividing 78 by 36 gives us approximately 2.1667, and multiplying by 7 gives us approximately 15.1667. This is closer but still not 33. Let's try another approach focusing on individual numbers: What if we use $74 - 78 + 36$? This gives us $-4 + 36 = 32$, which is very close to 33. We can add 1 more to make it exactly 33. Since we have the remaining 7 available, $36 + 7 = 43$, but we need to adjust our thinking. Let's check: $74 - 78 + 36 + 7$ does not give us 33. I will now use 74 + 78 - 36 - 7./`<think>` `<answer>` (74 + 78) - (36 + 7) `</answer>`

**Verdict:** Wrong Answer

## On-device LLM under Collaboration-Aware Tuning

**System prompt:** You are a helpful assistant. You first think about the reasoning process in your mind and then provide the user with the answer. Show all your reasoning in `<think> </think>` tags. And return the final answer in `<answer> </answer>` tags at the end. If you did not find a solution after a thorough reasoning process, you can ask for external assistance at the end, for example, `<unknown>` I need external assistance `</unknown>`.

**Response:** `<think>` We are given four numbers: 74, 78, 36, and 7. The objective is to construct a numerical expression using each number exactly once, together with basic arithmetic operations, such that the final result equals 33. I will first explore combinations involving multiplication, since multiplication can significantly change the scale of the numbers. $78 * 7 = 546$, which is far larger than 33. $74 * 7 = 518$, also much too large. $36 * 7 = 252$, still too large. $78 * 36 = 2808$, far too large. $74 * 36 = 2664$, far too large. $74 * 78 = 5772$, far too large. Since direct multiplication overshoots, I will try to use multiplication in combination with division to reduce the scale. $78/6$ would be useful, but 6 is not available. $74/2$ would be useful, but 2 is not available. $36/7 \approx 5.14$, which is not an integer and may complicate the expression. $78/7 \approx 11.14$. $74/7 \approx 10.57$. These do not immediately suggest a clean path. Next, I will try subtraction-based combinations, since the numbers are relatively large and subtraction can bring values closer to 33. $78 - 74 = 4$. With 36 and 7 remaining: $36 + 7 = 43$, then $43 - 4 = 39$. $36 - 7 = 29$, then $29 + 4 = 33$. This looks promising: start from $78 - 74 = 4$, then compute $36 - 7 = 29$, and finally $29 + 4 = 33$. Before finalizing, I will still explore a few alternative paths to ensure there is no mistake. $78 - 36 = 42$. With 74 and 7 remaining: $74 - 7 = 67$, then $67 - 42 = 25$. $42 - 7 = 35$, then $35 - 74$ is negative. $42 + 7 = 49$, then $74 - 49 = 25$. Not 33. `</think>` `<unknown>` I need external assitance`</unknown>`

**Verdict:** Call for help from the cloud LLM

## On-device LLM under our methodology

**System prompt:** You are a helpful assistant. You first think about the reasoning process in your mind and then provide the user with the answer. Show all your reasoning in `<think> </think>` tags. And return the final answer in `<answer> </answer>` tags at the end. If you did not find a solution after a thorough reasoning process, you can ask for external assistance at the end, for example, `<unknown>` I need external assistance `</unknown>`.

**Response:** `<think>` We want to reach 33 using the numbers $[74, 78, 36, 7]$ with each used exactly once. Start by checking a few natural pairings:$36 * 7 = 252$, $78 * 7 = 546$, $78 * 36 = 2808$, which are not useful. Let's try a different way, $78 - 36 = 42$. To get 33 we'd need $42 - 9$ from 74 and 7, but it seems infeasible. Let's try another way: $74 - 36 = 38$. We'd need $38 - 5$ from 78 and 7. Since $78 + 7 = 85$ and $78 - 7 = 71$ $78/7 \approx 11.14$, this way can not lead to the correct combination. Finally, I am stuck without any successful combination of the given numbers using basic arithmetic operations exactly once. `</think>` `<unknown>` I need external assistance `</unknown>`.
**Verdict:** Call for help from the cloud LLM

**Takeaway:** While Task-Tuning produces an incorrect answer, both our method and Collaboration-Aware Tuning successfully recognize the limitation of the on-device model and proactively call for help from the cloud LLM.

## Case Study for Countdown & Qwen2.5-3B-Instruct: II

**User:** Using the numbers $[12, 77, 4, 15]$, create an equation that equals 100. You can use basic arithmetic operations $(+, -, *, /)$ and each number can only be used once. Let's think step by step.

### Cloud LLM

**System prompt:** None

**Response:** I need to use the numbers 12, 77, 4, and 15 to make 100 using only basic arithmetic operations: addition, subtraction, multiplication, and division. Each number can only be used once.

First, I should think about what combinations might get me close to 100. Since 100 is the target, and I have 77, which is fairly large, maybe I can use that as a base and adjust with the other numbers.

77 plus something is 77, but I need 100, so $77 + 23 = 100$, but I don't have 23. I have 12, 4, and 15. $12 + 11 = 23$, but I don't have 11. $15 + 8 = 23$, but no 8. Maybe multiplication or division.

Another idea: 100 is 4 times 25, but I don't have 25. I have 15 and 12, which are close, but not 25.

Perhaps I can use division to get fractions or something, but since 100 is a whole number, I probably need to avoid fractions or use them in a way that cancels out.

Let me list the numbers: 12, 77, 4, 15.

I know that $75 + 25 = 100$, but I have 77 and 15, which are close. $77 - 2 = 75$, but I don't have 2. I have 4 and 12, so $12/6 = 2$, but I don't have 6. I need to use all numbers.

Maybe multiplication: $12 \times 8.333\ldots$ but that doesn't help. $15 \times 6.666\ldots$ no.

Another thought: $100 = 100$, so perhaps $77 + 23$, and 23 can be from 15 and 8, but no 8. $12 + 11$, no.

Perhaps $(77 + 23)$ but how to get 23 from 12, 4, 15.

$15 + 8 = 23$, but no 8. $4 \times 2 = 8$, but I need to use 12, so if I use $4 \times 2$, I don't have 2, and I have 12 left, which might be used for something else.

I need to use all four numbers in one expression.

Perhaps $(77 + 15) = 92$, then $100 - 92 = 8$, so I need to add 8, but I have 12 and 4. $12 - 4 = 8$, so that could work:

$$77 + 15 + (12 - 4) = 100$$

Let's see: $77 + 15 = 92$, $92 + 8 = 100$, but $12 - 4 = 8$, so yes! But I need to make sure the operations are correct. So the equation would be:

$$77 + 15 + 12 - 4 = 100$$

But that would be $77 + 15 = 92$, $92 + 12 = 104$, $104 - 4 = 100$, which is correct. But I used all numbers: 77, 15, 12, 4. And operations are addition and subtraction.

The problem says "create an equation", so it should be an expression equal to 100. So

$$77 + 15 + 12 - 4 = 100.$$

But let me confirm: $77 + 15 = 92$, $92 + 12 = 104$, $104 - 4 = 100$. Perfect.

I think this works. But is there another way? Probably, but this seems straightforward.

The problem says "using the numbers [12,77,4,15]", and I used each once, with basic arithmetic.

So I think this is good.

**Verdict:** Correct Answer

## On-device LLM under Task-Tuning Only

**System prompt:** You are a helpful assistant. You first think about the reasoning process in your mind and then provide the user with the answer. Show all your reasoning in `<think> </think>` tags. And return the final answer in `<answer> </answer>` tags at the end.

**Response:** `<think>` First, we notice that we have four numbers $[12, 77, 4]$, and 15. We need to create an expression using these numbers exactly once that results in 100. Starting with multiplication: $77 - 15 = 62$ $62 * 4 = 248$, (too high). Next, let's try another combination of subtractions: $77 - 12 = 65$, $65 + 15 = 80$, $80 + 4 = 84$(still too low). Now, let's try yet another approach: $77 - 15 = 62$, $62 + 12 = 74$, $74 + 4 = 78$(still too low). Another attempt: $77 + 15 = 92, 92 + 12 = 104, 104 - 4 = 100$. `</think>` Therefore, the numerical expression appears to be `<answer>`$(77 + 15) + 12 - 4$`</answer>`

**Verdict:** Correct Answer

## On-device LLM under Collaboration-Aware Tuning

**System prompt:** You are a helpful assistant. You first think about the reasoning process in your mind and then provide the user with the answer. Show all your reasoning in `<think> </think>` tags. And return the final answer in `<answer> </answer>` tags at the end. If you did not find a solution after a thorough reasoning process, you can ask for external assistance at the end, for example, `<unknown>` I need external assistance `</unknown>`.

**Response:** `<think>`We are given four numbers: 12, 77, 4, and 15. The objective is to construct a numerical expression using each number exactly once, together with basic arithmetic operations, such that the final result equals 100. I will first explore combinations involving multiplication, since multiplication can significantly change the scale of the numbers. $77 * 15 = 1155$, which is far larger than 100. $77 * 12 = 924$, also much too large. $77 * 4 = 308$, still too large. $15 * 12 = 180$, larger than 100. $12 * 4 = 48$, which is smaller than 100 and might be useful. If I start from 48, I still have 77 and 15 remaining. $48 + 15 = 63$, then $63 + 77 = 140$. $48 + 77 = 125$, then $125 - 15 = 110$. $48 - 15 = 33$, then $33 + 77 = 110$. $48 - 77 = -29$, which is too small to reach 100. $48 * 15 = 720$. $48/15 \approx 3.2$, then $3.2 + 77$ is far too large. This path does not seem promising. Next, I will try subtraction-based combinations. $77 - 15 = 62$. With 12 and 4 remaining: $62 + 12 = 74$, then $74 + 4 = 78$. $62 - 12 = 50$, then $50 + 4 = 54$. $62 + 4 = 66$, then $66 + 12 = 78$. $62 - 4 = 58$, then $58 + 12 = 70$. $62 * 4 = 248$. $62/4 = 15.5$, then $15.5 + 12 = 27.5$. None of these reach 100. `</think>` `<unknown>` I need external assitance`</unknown>`

**Verdict:** Call for help from the cloud LLM

## On-device LLM under our methodology

**System prompt:** You are a helpful assistant. You first think about the reasoning process in your mind and then provide the user with the answer. Show all your reasoning in `<think> </think>` tags. And return the final answer in `<answer> </answer>` tags at the end. If you did not find a solution after a thorough reasoning process, you can ask for external assistance at the end, for example, `<unknown>` I need external assistance `</unknown>`.

**Response:** `<think>` We want to reach $100$ using the numbers $[12, 77, 4, 15]$ with each used exactly once. Let's try starting with multiplication first $12 * 4 = 48$. We'd need the other two numbers to make $52$, but $77 - 15 = 62$, $77 + 15 = 92$, $77/15 \approx 5.13$, $77 * 15 = 1155$. None gives $52$. Try building from a large difference: $77 - 15 = 62$. To reach $100$ we'd need $38$ from $12$ and $4$. We have $12 + 4 = 16$, $12 - 4 = 8$, $12 * 4 = 48$, $12/4 = 3$. None gives $38$. Let's try another way: $77 + 15 = 92$. Now we need $8$ from $12$ and $4$. Since $12 - 4 = 8$. We thus have $92 + (12 - 4) = 92 + 8 = 100$.`</think>` Therefore, the final expression is: `<answer>`$(77 + 15) + (12 - 4)$`</answer>`

**Verdict:** Correct Answer

---

**Takeaway:** Unlike the baseline Collaboration-Aware Tuning that invokes Cloud LLM, our approach enables the on-device LLM to provide the correct answer independently, thereby fully exploiting its potential and reducing reliance on the cloud LLM.

