# OpenReview forum: "Bridging On-Device and Cloud LLMs for Collaborative Reasoning: A Unified Methodology for Local Routing and Post-Training"
_ICML.cc/2026/Conference — ICML 2026 regular_

### Official Review · Reviewer_Sog4 · 2026-02-27

**Soundness:** 2
**Presentation:** 3
**Significance:** 2
**Originality:** 3
**Overall Recommendation:** 3
**Confidence:** 4

**Summary:**

This work proposes a unified post-training framework for device-cloud collaborative reasoning, addressing the limitations of conventional external routers, which struggle to assess complex reasoning difficulty and incur high engineering maintenance costs. The framework internalizes routing decisions as special help-request tokens generated by the on-device model. Through hierarchical rewards and the GAPG algorithm, the model updates parameters on complementary difficult samples. Experiments show that the method achieves an optimal balance between independent on-device problem solving and invoking cloud assistance, improving reasoning accuracy while substantially reducing wasted local token computations.

**Compliance With Llm Reviewing Policy:**

Affirmed.

**Final Justification:**

The authors' rebuttal addressed most of my concerns, and I have raised my score. However, I believe that the technical approach of the router in this paper is not particularly novel, and it does not present significant progress or differentiation compared to prior work. That said, the overall quality of the work is relatively solid, placing it around the borderline level.

**Key Questions For Authors:**

See weaknesses above.

**Limitations:**

No, the paper does not provide a dedicated discussion on the framework’s technical limitations for practical deployment.

**Strengths And Weaknesses:**

**Strengths**

1. The external routing decision is internalized as a special token generation process within the on-device model, eliminating the need for an independent external routing classifier and significantly reducing the deployment complexity and subsequent maintenance cost of device-cloud collaborative systems.
2. The proposed Group Adaptive Policy Gradient (GAPG) algorithm, combined with adaptive prompt filtering, provides a practical and deployable engineering solution that maximizes device-cloud reasoning gains under strict offloading budget constraints.
3. The rl-based post-training framework treats the cloud model as a black-box oracle. Extensive experiments show that the pipeline generalizes well across multiple on-device models, including LLaMA-3.2 and Qwen-2.5, while significantly reducing token computation overhead on the device.

### Weaknesses

**1. Critical Missing Baselines in LLM Routing**
A major flaw of this paper is the complete lack of comparison with existing LLM router baselines designed for collaborative reasoning. To substantiate the proposed method's effectiveness, the authors must compare it against recent and highly relevant works, including:

* *Division-of-thoughts: Harnessing hybrid language model synergy for efficient on-device agents*
* *Router-r1: Teaching llms multi-round routing and aggregation via reinforcement learning*
* *Route-and-reason: Scaling large language model reasoning with reinforced model router*
* *Data shunt: Collaboration of small and large models for lower costs and better performance*
* *Automix: Automatically mixing language models*

Specifically, **Router-R1** is an essential baseline that must be included and compared against in the experiments.

**2. Unsupported Claims in the Introduction**
Two core claims made in the Introduction lack empirical evidence and factual basis. These claims form a key motivation for the paper and require supporting experimental evidence. But the corresponding experiments are also missing.

* **Claim 1:** *"Additionally, for reasoning tasks, the surface-level prompt features available to the local router often fail to reflect the underlying complexity of the problem."* Why? The paper provides no experimental evidence or empirical analysis to support the assertion that surface-level features are insufficient for local routers to gauge problem complexity.

* **Claim 2:** *"Such routers require extra task-specific training after the on-device LLM’s fine-tuning and make routing decisions solely based on the prompt, which limits generalization and adaptability across tasks."*
  The proposed methodology requires fine-tuning the on-device model itself, which introduces its own significant deployment inconveniences. From a system engineering perspective, fine-tuning the base model for routing might actually offer *worse* generalization and flexibility compared to maintaining a lightweight, decoupled external router. Furthermore, the paper lacks any rigorous evaluation of the on-device computational overhead and training costs.
  Finally, if the on-device model or the cloud model needs to be upgraded or swapped, can this tightly coupled framework easily transfer and generalize? The manuscript fails to discuss or evaluate this critical aspect of system adaptability.

**3. Risk of Premature Surrender and Shortcut Learning (Reward Hacking)**
RL is notoriously susceptible to reward hacking. Because the on-device model is guaranteed a baseline coordination reward ($\alpha_c$) as long as it outputs the `<unknown>` token and the cloud model answers correctly, it can easily converge on a sub-optimal policy. Specifically, the model might learn to immediately output `<unknown>` based merely on surface-level features in the prompt (e.g., the presence of complex vocabulary or lengthy context), rather than genuinely attempting intermediate reasoning and exiting only when it encounters a cognitive bottleneck. Although the authors claim the model learns to "self-assess its capabilities," the paper lacks a rigorous quantitative analysis of the on-device CoT traces. Without this analysis, it is impossible to rule out the possibility that the on-device model has simply degenerated into a shallow classifier that prematurely surrenders upon seeing a superficially difficult question.

---

> ### Author Rebuttal · Authors · 2026-03-29
>
> We sincerely thank the reviewer for the time, effort, and valuable feedback.
> ### **W1. More Baselines**
> To address this comment, we will add the following **discussion, references** and **experiments** to the revised manuscript.
>
> **Discussion.** Data Shunt routes queries using small-model confidence and relies on additional confidence distillation to improve the small model. AutoMix lets the small model answer and self-verify before deciding whether to keep its output or offload. Division-of-Thoughts decomposes a query into dependent sub-tasks and assigns them to either a local model or a cloud LLM. Recently, Route-and-Reason extends this line by studying sub-task decomposition and routing across multiple local and cloud models, while Router-R1 learns a reasoning-based router for centralized multi-agent settings via RL.
>
> **Experiments.** Among these works, **Division-of-Thoughts** and **Route-and-Reason** perform routing at the sub-task level, which differs from our task-level routing setup. We thus mainly compare our method with **Router-R1** and **AutoMix**. Note that Router-R1 employs a **separate** reasoning model for routing. In our comparison below, we use a router with the same size as the local model:
>
> |Model|Method|MATH-lighteval|MATH-500|AMC23|MinervaMath|AGI-Eval-Math|Avg.|
> |-|-|-:|-:|-:|-:|-:|-:|
> |Qwen2.5-1.5B|AutoMix|59.4|57.2|42.5|23.2|54.9|47.4|
> ||Router-R1|73.6|74.1|52.5|37.8|70.4|61.9|
> ||**Ours**|80.4|81.6|57.5|40.8|73.4|66.7|
> |LLaMA-3.2-3B|AutoMix|55.3|47.9|35.0|32.7|50.1|44.2|
> ||Router-R1|72.7|64.9|47.5|39.7|59.2|56.8|
> ||**Ours**|79.5|68.6|52.5|43.4|64.5|61.7|
>
> > Compared with using an external model for routing, ours with internal routing achieves a notably higher overall accuracy.
>
> ### **W2-1. Empirical evidence for Claims 1 and 2**
> We apologize for any ambiguity in our original phrasing. These statements are intended to refer specifically to lightweight routers without reasoning capability, rather than to all LLM routing methods. In our device-cloud collaborative setting, using a separate reasoning-capable router (e.g.,Router-R1) would introduce additional local memory overhead. We studied the limitations of a prompt-only classifier in **Appendix A.4** (Figure 12). Moreover, in **Table 1** and **Figure 4**, Task-Tuning&Router consistently underperforms our method. In particular, **Table 1** evaluates both methods on multiple unseen tasks after training solely on MATH-lighteval. Our method's consistent advantage over this external-router baseline in that setting further indicates stronger **cross-task generalization**.
>
> > We will make this scope explicit in the revised manuscript and highlight these supporting results.
>
> ### **W2-2. Tuning the on-device model for routing introduces significant deployment inconveniences & Model change**
> We would like to clarify that our method does **not** require a separate procedure to learn routing. Instead, routing is learned during the **same** task-specific post-training stage used to train the model's task-solving ability. Since all baselines require post-training the on-device LLM, this does not introduce an additional deployment cost.
>
> Our framework is **not** tied to a specific on-device model instance. Before deployment, a new on-device model can naturally learn routing during its task-specific post-training stage under our framework.
>
> `We added an experiments on cloud-model swapping in our response to Reviewer QUTs, W2 & Q1. Due to space limits, we refer the reviewer there for the results.`
>
> ### **W2-3. On-device computational overhead and training costs**
> `We have added a discussion and results in our response to Reviewer krpF, W2. Due to space limits, we refer the reviewer there for the details.`
>
> ### **W3. Risk of Premature Surrender**
> **Coordination reward hacking.** We explored this in **Appendix A.1.2**. When the coordination reward is set too high, the model can indeed collapse. Importantly, **Figures 7 and 8** show that within a reasonably broad range of coordination reward values ($\\alpha_c$), the on-device model learns both task solving and routing behavior, without collapsing.
>
> **Quantitative CoT analysis.** We presented some results in **Figure 11**, which shows the lengths of **calling-for-help** responses for Llama-3.2-3B and Qwen2.5-1.5B, and **Appendix G**, which provides representative examples showing that the model performs intermediate reasoning before deciding to call for help.
>
>
> To further address this concern, we provide a quantitative analysis of the on-device CoT traces based on the  MATH-lighteval test set (5000 samples).
>
> |Metric|Qwen2.5-1.5B|Llama-3.2-3B|
> |-|-:|-:|
> |Help traces|1607|1471|
> |Mean steps pre-help|4.92|4.19|
> |Median steps pre-help|5|4|
> |Mean tokens pre-help|185|113|
> |Median tokens pre-help|179|98|
> |Asking help within 30 tokens|0.37%|0.59%|
>
> > These CoT-trace statistics show that the on-device model **does not** behave like a shallow classifier. We will add this analysis to Appendix A.3.

---

> > ### Author Rebuttal · Reviewer_Sog4 · 2026-04-03
> >
> > The rebuttal has resolved most of my initial concerns. I will increase my score from 2 to 3. I hope the authors to include all the clarifications and new experiments from the rebuttal period into the final paper.

---

> > > ### Author Response · Authors · 2026-04-04
> > >
> > > Thank you very much for your acknowledgment. We are encouraged that our response addressed most of your concerns, and we sincerely appreciate your decision to raise the score.
> > >
> > > In the revision, we will incorporate discussion of all these suggested methods into Related Work (Sec. 1.2), include the new experiments in Table 1 (Sec. 4.1), clarify the scope of our claims in the Introduction (Sec. 1) while highlighting the supporting results, and add new quantitative analysis of CoT traces to Appendix A.3.
> > >
> > > If there are any remaining concerns or further suggestions that could make you more positive toward our work, we would be eager to address them to the best of our ability.

---

### Official Review · Reviewer_QUTs · 2026-03-13

**Soundness:** 3
**Presentation:** 3
**Significance:** 3
**Originality:** 3
**Overall Recommendation:** 5
**Confidence:** 3

**Summary:**

The paper proposes folding cloud-routing decisions directly into the on-device LLM's RL post-training, eliminating the need for a separate external router. The core idea is that instead of training a classifier to predict query difficulty from prompt features (which works poorly for reasoning tasks), you train the on-device model to recognize its own limitations and emit a "call for help" token when stuck. The method includes the following:

- A hierarchical reward scheme that gives higher reward for correct local answers than correct cloud-assisted answers, discouraging over-reliance on offloading
- A Group-Adaptive Policy Gradient (GAPG) algorithm with a provably unbiased, variance-reduced gradient estimator
- An adaptive prompt filtering mechanism that keeps training balanced between locally-solvable and cloud-requiring prompts, preventing the policy from collapsing to always calling for help

Experiments across four on-device models (1B to 3B) on math and symbolic reasoning benchmarks show consistent gains over both naive offloading and trained router baselines, under a fixed 30% cloud call budget.

**Compliance With Llm Reviewing Policy:**

Affirmed.

**Final Justification:**

The rebuttals addressed my concerns, and I have decided to raise my score.

**Key Questions For Authors:**

1. **How sensitive is the method to the choice of cloud model?** All experiments use DeepSeek-R1. If the cloud model were weaker or swapped out at inference time, would the routing behavior learned during training still be appropriate?

2. **How should practitioners set αc for a new model family?** The ablations show the optimal range shifts substantially between Qwen2.5-1.5B and Llama-3.2-3B with no overlap. Is there a principled way to pick this value without running the full sweep, or does it always require a grid search? A clear answer would address the reproducibility concern.

**Limitations:**

Yes

**Strengths And Weaknesses:**

**Strengths:**

- The core insight is clean and well-motivated: replacing an external binary router with the on-device model's own learned self-awareness sidesteps the fundamental problem that prompt surface features don't reliably predict reasoning difficulty. The case studies in Appendix H make this concrete in a way that's easy to follow.
- The theoretical grounding is solid. The proof that GAPG produces an unbiased gradient estimator with variance decaying as O(1/G) is straightforward and the group size ablation (Table 2, Figures 5-6) confirms the theory empirically. The Lagrangian relaxation derivation in Appendix B also gives a principled justification for why the hierarchical reward structure naturally falls out of the constrained optimization problem.
- The ablations are honest and informative. Figure 14 showing GRPO collapsing to a 100% call-for-cloud policy is a compelling negative result that genuinely motivates the GAPG design. The reward weight sweep in Figures 7-8 also clearly shows the failure modes on both ends.
- The prompt filtering mechanism is well-justified both theoretically (Appendix B, equation 15) and empirically (Figure 9), and the failure mode without it is shown directly rather than just described.


**Weaknesses:**

- The evaluation is limited to math and symbolic reasoning tasks where correctness-based rewards are easy to verify. The paper acknowledges this in the limitations section, but it means the applicability of the approach to more general tasks (summarization, instruction following, code) is completely open. Given that this is positioned as a general device-cloud collaboration framework, that's a notable gap.
- The cloud model is always DeepSeek-R1. There's no experiment varying the capability gap between the on-device and cloud models, which matters a lot in practice. It's unclear whether the learned routing behavior would transfer if the cloud model changed
- The call-for-help behavior shown in Appendix G is interesting but only covers Qwen2.5-1.5B on MATH-lighteval. It would be more convincing to see whether the early-exit routing patterns generalize across models and tasks, rather than presenting a single qualitative case study.
- The baseline router (DeBERTa-v3-large) is relatively weak. Comparing against a stronger routing baseline, such as a small LLM-based router or a confidence-calibrated approach, would make the advantage of internal routing more convincing.

---

> ### Author Rebuttal · Authors · 2026-03-29
>
> We sincerely thank the reviewer for the time, effort, and valuable feedback.
> ### **W1. Applicability to more general tasks**
> This work indeed focuses on collaborative reasoning, a commonly encountered setting where verifiable rewards are available [R1, R2]. Our goal is to establish the feasibility of device-cloud collaboration via end-to-end learned help-seeking behavior in the reasoning domain. Math and symbolic reasoning therefore serve as a controlled testbed for this purpose.
>
> We view our work as a strong complement to existing router-based methods in the reasoning domain. We agree with the reviewer that extending the framework to broader domains, including summarization and other open-ended tasks, is an important direction for future work as we acknowledge in our limitations.
>
> [R1] Guo, Daya, et al. "Deepseek-r1: Incentivizing reasoning capability in llms via reinforcement learning." 2025
>
> [R2] Jin, Bowen, et al. "Controlling Performance and Budget of a Centralized Multi-agent LLM System with Reinforcement Learning." 2025
>
> > In addition to in the title (**collaborative reasoning**) and limitations section, we will further clarify our focus on reasoning tasks in the introduction.
>
> ### **W2 & Q1. Switching cloud LLM at inference time**
> To address this, we evaluate cloud-model swapping at inference time by replacing the original cloud model (**DeepSeek-R1**) with a weaker alternative (**Qwen3-Qwen3-30B-A3B-Instruct**). We then test our method with Qwen2.5-1.5B as the local model under both cloud choices (offloading ratio 30%):
>
> |Setting|CloudLLM|MATH-lighteval|MATH-500|AMC23|MinervaMath|AGI-Eval-Math|Avg.|
> |-|-|-:|-:|-:|-:|-:|-:|
> |Cloud only|DeepSeek-R1|98.4|97.3|97.5|80.9|94.7|93.8|
> |Cloud only|Qwen3-30B-A3B-Instruct|95.6|93.8|97.5|77.6|93.3|91.6|
> |Ours|DeepSeek-R1|80.4|81.6|57.5|40.8|73.4|66.7|
> |Ours|Qwen3-30B-A3B-Instruct|79.5|80.4|57.5|40.1|73.2|66.1|
>
> > The learned routing policy mainly captures when local capability is insufficient and cloud assistance should be requested. Therefore, changing the cloud LLM at inference time has limited impact as long as the new cloud model remains substantially stronger than the on-device model.
>
> ### **W3. Call-for-help behavior in Appendix G**
>
> In Appendix A.3 (**Figure 11**), we showed that for both Llama-3.2-3B and Qwen2.5-1.5B, the lengths of calling-for-help responses are shorter than those of independent reasoning responses. This suggests that the early-exit behavior appears across different models.
>
> To show this phenomenon across tasks, we further report the lengths of **calling-for-help** and **independent reasoning** responses for Llama-3.2-3B across tasks below.
>
> | Type | MATH-lighteval | MATH-500 | AMC23 | MinervaMath | AGI-Eval-Math | Avg. |
> |-|-:|-:|-:|-:|-:|-:|
> | Independent reasoning | 327 | 385 | 402 | 582 | 453 | 430 |
> | Calling for help | 109 | 181 | 195 | 253 | 289 | 205 |
>
> > We see that the early-exit call-for-help pattern appears consistently across different models and tasks. We will add this table to Appendix G.
>
> ### **W4. More baselines**
>
> We have conducted new experiments with **Router-R1** and **AutoMix** as baselines. Router-R1 uses an extra reasoning model for routing; in our setup, we instantiate it with the same base model as the on-device LLM. AutoMix allows the local model to first answer and self-verify (2 pass), and then decide whether to keep the local output or offload to the cloud.
>
> | Model | Method | MATH-lighteval | MATH-500 | AMC23 | MinervaMath | AGI-Eval-Math | Avg. |
> | - | - | -: | -: |-: |-: |-: |-:|
> | Qwen2.5-1.5B | AutoMix | 59.4 | 57.2 | 42.5 | 23.2 | 54.9 | 47.4 |
> || Router-R1 | 73.6 | 74.1 | 52.5 | 37.8 | 70.4 | 61.9 |
> || **Ours** | 80.4 | 81.6 | 57.5 | 40.8 | 73.4 | 66.7 |
> | LLaMA-3.2-3B | AutoMix | 55.3 | 47.9 | 35.0 | 32.7 | 50.1 | 44.2 |
> || Router-R1 | 72.7 | 64.9 | 47.5 | 39.7 | 59.2 | 56.8 |
> || **Ours** | 79.5 | 68.6 | 52.5 | 43.4 | 64.5 | 61.7 |
>
> > These results further support the advantage of our internal routing design over external router in reasoning settings. We will add these baselines in the manuscript.
>
> ### **Q2. How to set $\alpha_c$ for a new model family?**
>
> To select $\alpha_c$, we can leverage the observation that good performance is achieved over a **reasonably broad interval** of $\alpha_c$ rather than at a single sharp optimum. Concretely, in **Figure 7** (Qwen2.5-1.5B), strong performance is maintained for $\alpha_c \in [0.25, 0.45]$, while in **Figure 8** (Llama-3.2-3B), the stable high-performing region is $\alpha_c \in [0.55, 0.70]$. While the specific intervals may not overlap, it implies that for each model family, we can test a **small set of widely spaced $\alpha_c$ candidates** (e.g., \{0.2, 0.4, 0.6, 0.8\}), rather than conducting an exhaustive task-specific grid search. Inappropriate values are easy to identify because they typically cause **rapid early collapse** toward one-sided coordination behavior, visible early in training.
>
> > We will add this discussion to **Appendix A.1.2**.

---

> > ### Author Rebuttal · Reviewer_QUTs · 2026-03-31
> >
> > Thanks for addressing my concerns. In response, I would like to raise my score to a 5 (accept).

---

> > > ### Author Response · Authors · 2026-04-01
> > >
> > > Thank you very much for your kind acknowledgment and for letting us know that our response addressed your concerns. We truly appreciate your time and consideration.
> > >
> > > We also noticed that the score shown in the system may still be the previous one. We would be very grateful if you could update it to match your latest assessment.
> > >
> > > Thank you again for your support.

---

### Official Review · Reviewer_6zbU · 2026-03-13

**Soundness:** 3
**Presentation:** 3
**Significance:** 2
**Originality:** 3
**Overall Recommendation:** 4
**Confidence:** 3

**Summary:**

The paper introduces a unified RL method for device-cloud LLM collaboration. Instead of relying on a decoupled, external classifier to determine task difficulty, the authors train the on-device model to autonomously decide when to generate a local answer versus when to offload to a more capable cloud model. They frame this as a reward maximization problem subject to cloud usage constraints, introducing a Group-Adaptive Policy Gradient algorithm and a hierarchical reward structure to stabilize the joint learning of reasoning and routing.

**Compliance With Llm Reviewing Policy:**

Affirmed.

**Final Justification:**

The rebuttal addressed my main concerns and improved my confidence in the paper. However, it does not substantially change my assessment of the paper's overall significance and novelty. I will keep my current score.

**Key Questions For Authors:**

see weakness

**Limitations:**

yes

**Strengths And Weaknesses:**

### Strengths:
* **Methodological integration:** Integrating the routing decision directly into the RL post-training of the on-device model is a highly practical and elegant departure from standard cascaded or router-based pipelines. It treats routing as an intrinsic capability rather than a surface-level classification task.
* **Theoretical grounding:** The variance reduction properties of the group-level policy gradient estimator (Proposition 3.1) are sound, and the Lagrangian relaxation used to justify the hierarchical reward is well-reasoned.
* **Comprehensive baselines:** The empirical comparison against traditional Task-Tuning & Router baselines is strong, particularly showing that GAPG prevents the policy from collapsing into 100% cloud reliance compared with standard GRPO in Appendix.

### Weaknesses:
1. **Generalization to Diverse Model Space:** The current method strictly considers a binary routing decision: local execution vs offloading to a single monolithic cloud model. In real-world AI systems, we often need to route queries to a diverse subset of specialized models. Could the proposed GAPG algorithm and hierarchical reward structure be generalized to support subset selection among multiple heterogeneous cloud experts, or does the reward formulation restrict it to a 1:1 local-to-cloud mapping?
2. **Cold-Start:** The adaptive prompt filtering mechanism constructs a subset $\mathcal{D}_b^1$ consisting of prompts the local model can solve, and $\mathcal{D}_b^2$ where only the cloud model succeeds. If the initial on-device model is extremely weak relative to the task (i.e., it cannot solve any problems in the batch), $\mathcal{D}_b^1$ becomes empty. How does the algorithm handle this scenario to prevent early training collapse?
3. **Reward Weight Sensitivity:** Based on Appendix A.1.2, the method is quite sensitive to the ratio of the accuracy reward ($\alpha_a$) to the coordination reward ($\alpha_c$). A slight overtuning of $\alpha_c$ causes the model to collapse into constantly requesting help. How did you solve this? Does it require exhaustive, task-specific hyperparameter grid search?

---

> ### Author Rebuttal · Authors · 2026-03-29
>
> We sincerely thank the reviewer for their time, effort, and valuable feedback.
>
> ### **W1. Algorithm and hierarchical reward structure generalize to multi-cloud setup**
> In this work, we consider single-model selection per query. We hesitate to extrapolate our current work to selecting multiple LLMs for one query, because such decentralized LLM coordination is challenging for capacity-limited local models. Nonetheless, our proposed hierarchical reward structure **can be naturally generalized to a multi-cloud setup** in which one of multiple cloud models (or the local model) is selected per query, following a similar derivation as in Appendix B. We explain this extension below.
>
> **Multi-cloud hierarchical reward.** Suppose the on-device model can offload an input to one of $K$ cloud models. Let $\mathbb{1}_k(y)$ denote the indicator that the final answer for sample $x$ is obtained using cloud $k$. We consider the constrained optimization
>
> $$
> \max_{\theta} \mathbb{E}[r(x,y)]
> \quad
> \text{s.t.}~~
> \mathbb{E}[\mathbb{1}_k(y)] \le \rho_k,
> \ \forall k.
> $$
>
> Lagrangian relaxation with multipliers $\gamma_k \ge 0$ yields
>
> $$\mathcal{L}(\theta,\gamma)=\mathbb{E}[r(x,y)] - \sum_{k=1}^K \gamma_k (\mathbb{E}[\mathbb{1}\_k(y)] - \rho\_k ).$$
>
> Notably, $\gamma_k$ is treated as a hyperparameter. Dropping the constant term $\sum_{k=1}^K \gamma_k \rho_k$, the objective becomes
>
> $$
> \max_{\theta}\ \mathbb{E}\left[
> r(x,y) - \sum_{k=1}^K \gamma_k \mathbb{1}_k(y)
> \right].
> $$
>
> Therefore, the constrained problem is equivalent to optimizing the modified reward
>
> $$
> \\tilde r(x,y)=
> \\begin{cases}
> r(x,y)-\\gamma_k, & \\text{for cloud } k, \\\\
> r(x,y), & \\text{local}.
> \\end{cases}
> $$
>
> Thus, different cloud models induce different coordination penalties, yielding a multi-level hierarchical reward.
>
> > In the revision, we will clarify that our focus is on the 1:1 device-cloud setting, but that our framework can be extended to the multi-cloud setting in which one query is routed to one of many LLMs.
>
>
> ### **W2. Extremely weak on-device model & Empty $D_b^1$**
>
> We would like to clarify that our algorithm does not rely solely on the on-device model’s current pass@1 accuracy. We sample a group of $G$ responses for each query, and a sample is included in $D_b^1$ as long as at least one response from the on-device model is correct. Therefore, even when pass@1 is low, $D_b^1$ is unlikely to be empty as long as the initial model has sufficiently strong **pass@$G$**. In practice, if $D_b^1$ becomes too small, a direct mitigation is to increase $G$, which raises the chance of obtaining at least one correct on-device response and thus enlarges $D_b^1$.
>
> More broadly, before conducting RL based post-training, one should first assess whether device-cloud collaboration is meaningful for the target training task by evaluating the on-device model’s initial **pass@$k$** (e.g., $k=4$). In our experiments, for example, the initial **pass@4** of both **Qwen2.5-1.5B** and **Llama-3.2-3B** exceeds **60%**, indicating a reasonably capable on-device model.
>
> If the model is extremely weak and **pass@$k$** is close to zero even with a large $k$ (e.g., 32), then $D_b^1$ may indeed become empty. However, this is **not a failure mode specific to our method**; rather, it reflects a broader limitation of on-policy RL post-training methods [R1], including the baselines considered in our paper, all of which rely on RL-based post-training. More fundamentally, it indicates that the task lies outside the effective operating regime of on-policy RL post-training and device-cloud collaboration.
>
> [R1] Yue, Yang, et al. "Does reinforcement learning really incentivize reasoning capacity in llms beyond the base model?." NeurIPS 2025
>
> > We will add a table of pass@$k$ for our experiments and explain the broader limitation of on-policy RL post-training.
>
> ### **W3. Reward weight sensitivity & Exhaustive grid search**
>
> In **Appendix A.1.2**, we see the model collapses toward constantly requesting help only when the coordination reward is set **too high**. Strong performance is achieved over a **reasonably broad interval** rather than at a single sharp optimum. Concretely, in **Figure 7** (Qwen2.5-1.5B), strong performance is maintained for $\alpha_c \in [0.25, 0.45]$, while in **Figure 8** (Llama-3.2-3B), the stable high-performing region is $\alpha_c \in [0.55, 0.70]$. Thus, the effective region spans multiple neighboring values in each case.
>
> Practically, we handle this by using a **small set of widely spaced $\alpha_c$ candidates** (e.g., \{0.2, 0.4, 0.6, 0.8\}) rather than an exhaustive task-specific grid search. Inappropriate values are easy to identify as they typically cause **rapid early collapse** toward one-sided coordination behavior, which becomes visible early in training. Therefore, for a new model, a coarse search over $(0,1)$ with early monitoring is a practical tuning strategy.
>
> > We will add this discussion into **Appendix A.1.2**.

---

> > ### Author Rebuttal · Reviewer_6zbU · 2026-04-03
> >
> > Thanks for the rebuttal. It addressed my main concerns and improved my confidence in the paper. However, it does not substantially change my assessment of the paper's overall significance and novelty. I will keep my current score.

---

> > > ### Author Response · Authors · 2026-04-04
> > >
> > > Thank you very much for your kind acknowledgment and for letting us know that our response addressed your concerns. We sincerely appreciate your time and consideration.

---

### Official Review · Reviewer_krpF · 2026-03-13

**Soundness:** 3
**Presentation:** 3
**Significance:** 3
**Originality:** 3
**Overall Recommendation:** 5
**Confidence:** 4

**Summary:**

This paper trains an on-device LLM to both answer queries locally and decide when to offload to a stronger cloud LLM under a call budget, instead of using a separate router. It formulates this as a constrained RL problem with hierarchical rewards and introduces a group-based policy gradient plus prompt filtering to stably learn both solving and routing behaviors. Experiments on math/symbolic reasoning show better accuracy–budget trade-offs than local-only, naive offload, router-based, and Dr.GRPO-style baselines.

**Compliance With Llm Reviewing Policy:**

Affirmed.

**Final Justification:**

My concerns have been addressed, and I will keep my original score.

**Key Questions For Authors:**

See weakness

**Limitations:**

Yes

**Strengths And Weaknesses:**

Strengths:

1.	Routing is built directly into the on-device model's RL post-training, so there's no need for a separate router. For real systems, this is a pretty sensible design choice.
2.	Compared to GRPO's empirically designed group loss, the proposed group-level estimator is explicitly derived as an unbiased policy gradient for the constrained objective, with a proven O(1/G) variance bound. This gives the method a stronger theoretical basis in the device-cloud setting.


Weakness:

1.	The method relies heavily on verifiable rewards. The experiments focus on math and symbolic reasoning tasks where correctness can be checked automatically with rules. For more general collaborative inference scenarios like open-ended QA, dialogue, or code, it's unclear how to build a reliable reward signal.
2.	Device-cloud setups inherently care a lot about latency, token cost, and on-device energy use. But this paper mostly just talks about cloud call ratio and accuracy in the main text. The deeper cost analysis is pushed to the appendix, and even there it only covers token and call counts. There's not enough discussion of the system-level metrics that actually matter in real deployment.

---

> ### Author Rebuttal · Authors · 2026-03-29
>
> We thank the reviewer for their time, effort, and positive feedback.
>
> ### **W1. More general collaborative inference scenarios**
>
> Our focus in this work is on reasoning settings where verifiable rewards are typically available, e.g., as in [R1, R2]. We attempted to highlight this upfront, including in the title (**collaborative reasoning**), as well as in the limitations section, given the reviewer's point that non-reasoning scenarios without clear reward signals are not easily applicable to our current framework. Our goal in this work is to establish the feasibility of device-cloud collaboration via end-to-end learned help-seeking behavior in the reasoning domain. Math and symbolic reasoning therefore serve as a controlled testbed for this purpose.
>
> We agree that extending the framework to more general collaborative inference scenarios, such as open-ended QA, is an important next step. In such settings, reward assignment is indeed more challenging because correctness is less directly observable. That said, our current work should be viewed as a **complement** to existing router-based approaches in the **reasoning domain**, where verifiable rewards are available and where external routers often struggle. We see reward design for open-ended tasks as an important future direction. We will further emphasize this in the introduction and in the limitation section.
>
> [R1] Guo, Daya, et al. "Deepseek-r1: Incentivizing reasoning capability in llms via reinforcement learning." 2025
>
> [R2] Jin, Bowen, et al. "Controlling Performance and Budget of a Centralized Multi-agent LLM System with Reinforcement Learning." 2025
>
> ### **W2. Cost Analysis, System-level metrics like latency, token cost, and on-device energy use**
>
> We agree with the reviewer that latency, token cost, and on-device energy are important deployment metrics. We provided partial analysis in the appendix, although this may not have been sufficiently highlighted in the main paper:
>
> **Cost Analysis.** Appendix A.2 compares **training cost** in terms of cloud LLM usage. Since all compared methods rely on RL-based fine-tuning, their policy-update complexity is similar, and the main difference lies in cloud LLM usage. Moreover, the number of RL post-training steps is matched across our method and the baselines (see **Figures 2 and 3**), so the required GPU hours are comparable. Importantly, this cost is incurred **offline** during post-training, prior to deployment, rather than on-device during inference. Appendix A.3 reports **on-device token efficiency**, including the response length of independent reasoning and call-for-help behaviors.
>
> **Latency and on-device computational overhead/energy.** Latency in our setting consists of three components: local computation latency, device-cloud communication latency, and cloud computation latency. Since the offloading ratios are set to be the same across baselines, the communication latency and cloud computation latency will be comparable across these methods.
> Therefore, we focus on the **local computation latency**, which is directly determined by the amount of on-device token generation. Similarly, while we do not measure hardware power directly in this work, on-device **energy** use and on-device computational overhead are also closely tied to local computation and can be approximated by **local token generation**.
>
> We compare on-device token cost and **estimated local latency** under Llama-3.2-3B on the MATH-lighteval test set (5000 samples). The latency is estimated from a decoding speed of **65.83 tokens/s** reported by **NVIDIA Jetson AI Lab** for **Jetson Thor** (jetson-ai-lab.com/models/llama3-2-3b/).
>
> | Method | On-device token cost per query | Estimated local latency (s) |
> | - | -: | -: |
> | Task Tuning Only | 357 | 5.42 |
> | Task-Tuning&Router | 213 | 3.24 |
> | Random Offloading | 250 | 3.80 |
> | Collaboration-Aware Tuning | 302 | 4.59 |
> | AutoMix | 779 | 11.83 |
> | Router-R1 | 381 | 5.79 |
> | **Ours** | 251 | 3.81 |
>
> Router-R1 and AutoMix are two baselines added during the rebuttal stage (see our response to **Reviewer QUTs**, W4). Router-R1 uses an **extra reasoning model** for routing; in our setup, we instantiate it with the same base model as the on-device LLM. AutoMix lets the local model first answer and then self-verify (**two passes**) before deciding whether to keep the local output or offload to the cloud.
>
> From the above table, we see that our method incurs slightly higher local token cost and local latency than Task-Tuning&Router, but remains more efficient than AutoMix and Router-R1, while achieving significantly better routing quality and overall accuracy (as seen in our other results).
>
> > We will further emphasize the discussion on on-device computational overhead in Appendix A.3, explicitly clarify that local tokens serve as a proxy for local latency and energy, and add the latency comparison experiments into the revised manuscript.

---

> > ### Author Rebuttal · Reviewer_krpF · 2026-04-03
> >
> > Some of our questions were resolved, so I maintain my score.

---

> > > ### Author Response · Authors · 2026-04-04
> > >
> > > Thank you very much for your kind acknowledgment and for letting us know that our response addressed your concerns. We sincerely appreciate your time, consideration, and support.

---

### Decision · Program_Chairs · 2026-04-30

**Decision:**

Accept (regular)

**Comment:**

The reviewers are in general agreement that this paper presents a well-motivated approach to device-cloud collaborative reasoning by internalizing routing into the on-device model's RL post-training, with solid theoretical grounding. The main concerns—missing strong routing baselines, system-level cost analysis, and risk of reward hacking—were substantively addressed during the rebuttal with new experiments and quantitative analysis. Three reviewers are positive (5, 4, 5), and the remaining reviewer raised their score acknowledging that most concerns were resolved, with a residual concern on novelty that is not shared by the other reviewers. I recommend acceptance. The authors should incorporate the promised additional baselines and analyses into the camera-ready.